# The mediating role of systemic inflammation and moderating role of racialization in disparities in incident dementia

César Higgins Tejera [1,2] ✉, Erin B. Ware[3], Margaret T. Hicken[3], Lindsay C. Kobayashi[1], Herong Wang [1], Freida Blostein[4], Matthew Zawistowski[1], Bhramar Mukherjee[1] & Kelly M. Bakulski [1]

## Abstract

**Background** Exposure to systemic racism is linked to increased dementia burden. To assess systemic inflammation as a potential pathway linking exposure to racism and dementia disparities, we investigated the mediating role of C-reactive protein (CRP), a systemic inflammation marker, and the moderating role of the racialization process in incident dementia.

**Methods** In the US Health and Retirement Study (n = 6,908), serum CRP was measured at baseline (2006, 2008 waves). Incident dementia was classified by cognitive tests over a six-year follow-up. Self-reported racialized categories were a proxy for exposure to the racialization process. We decomposed racialized disparities in dementia incidence (non-Hispanic Black and/or Hispanic vs. non-Hispanic white) into 1) the mediated effect of CRP, 2) the moderated portion attributable to the interaction between racialized group membership and CRP, and 3) the controlled direct effect (other pathways through which racism operates).

**Results** The 6-year cumulative incidence of dementia is 12%. Among minoritized participants (i.e., non-Hispanic Black and/or Hispanic), high CRP levels (≥ 75th percentile or 4.73μg/mL) are associated with 1.26 (95%CI: 0.98, 1.62) times greater risk of incident dementia than low CRP (< 4.73μg/mL). Decomposition analysis comparing minoritized versus non-Hispanic white participants shows that the mediating effect of CRP accounts for 3% (95% CI: 0%, 6%) of the racial disparity, while the interaction effect between minoritized group status and high CRP accounts for 14% (95% CI: 1%, 27%) of the disparity. Findings are robust to potential violations of causal mediation assumptions.

**Conclusions** Minoritized group membership modifies the relationship between systemic inflammation and incident dementia.

## Plain language summary

Higher levels of inflammation in blood are linked to greater dementia risk in older adults. Non-Hispanic Black and Hispanic Americans have higher inflammation levels compared to non-Hispanic white Americans. We conducted a study to examine whether high levels of inflammation could explain differences in dementia risk among these racial groups. We found that differences in inflammation levels in non-Hispanic Black or Hispanic adults modestly explain their higher risk of dementia compared to non-Hispanic white adults. These findings suggest that interventions aimed at reducing high levels of inflammation in minoritized US adults could ameliorate racial differences in dementia risk.

Dementia is an important contributor to morbidity and mortality in the United States and a debilitating condition that requires caregiving and support with activities of daily living[1–3]. The burden of dementia is expected to increase due to demographic changes in the elderly population, lack of early diagnosis, and definite treatment[4]. Disparities in prevalence and incident dementia have been documented in the United States[5,6]. Non-Hispanic Black and Hispanic Americans are more likely to develop dementia than their white counterparts[5–7]. The increasing burden of dementia will disproportionately affect populations[8] made vulnerable by increased risk of developing other chronic conditions[9–12]. Historically, many studies linking race to health outcomes in medicine, public health, and epidemiology have wrongly concluded that such health disparities are attributable to ancestral or cultural differences

[1]School of Public Health, University of Michigan, 1415 Washington Heights, Ann Arbor, MI, 48109, USA. [2]Department of Neurology, Division of Neuroimmunology and Neurological Infections, Johns Hopkins University, 600 N Wolfe St, Baltimore, MD, 21287, USA. [3]Institute for Social Research, University of Michigan, 426 Thompson St, 48104 Ann Arbor, MI, USA. [4]Vanderbilt University, 2525 West End Avenue, 37203 Nashville, TN, USA. ✉e-mail: chiggi25@jh.edu

ascribed to these racialized groups[13–15]. Contrarily, historians, sociologists, and social epidemiologists argue disparities exist due to racialization, a process by which individuals are grouped into social categories (i.e., racialized groups) and where access to resources and opportunities are granted or denied[16–18]. Racialization, a key component of structural racism—the system that assigns a race to individuals and differentially provides political and socioeconomic resources to groups based on the value ascribed to the race—[16,19–21] is an important determinant of health disparities[22–24] as it differentially exposes groups to risk in ways that ultimately influence physiological responses and increase susceptibility to health and disease[16,25]. Theoretical frameworks such as the weathering hypothesis[19] and biological embedding[26] provide a blueprint of this pathway by describing how minoritized racialized groups experience deteriorated physiological function as a consequence of persistent marginalization, economic deprivation, and political under-representation. Therefore, disparities among racialized groups are the product of structural racism[25,27–30].

C-reactive protein (CRP), a marker of systemic inflammation, may capture the impact of racialization on physiological responses and consecutively in cognitive aging. The colocalization of glia and pro-inflammatory cytokines in amyloid-β plaques implicate that neuro-inflammation has an important role in the pathogenesis of dementia[31–33]. The neurodegenerative process that follows the extracellular deposits of amyloid-β peptide, the activation of the glial, and the production of pro-inflammatory cytokines suggest that inflammation may be the result of a reaction to the abnormal accumulation of proteins in brain parenchyma[31,34]. However, mounting evidence from observational epidemiological studies suggests a link between systemic inflammation and dementia onset[31,35–37]. Pro-inflammatory cytokines can induce permeabilization of the brain blood barrier endothelium, inducing paracrine signaling with surrounding macrophages, and activation of the microglia[33]. Therefore, increasing epidemiological evidence suggests that systemic inflammation may be a driving force in the chain of events that lead to the onset of dementia[31,32,36,38,39].

In addition, systemic inflammation may explain racialized disparities in cognitive aging. The weathering hypothesis proposes that structural racism regularly activates the body's stress-response causing minoritized racialized individuals to experience allostatic overload[19,20]. The hyper-activation of the hypothalamus-pituitary-axis may lead to a chronic stress response characterized by elevated biomarkers of systemic inflammation (i.e., CRP, interleukin-6, tumoral necrosis factor-alpha), and stress hormones (i.e., cortisol, adrenaline)[40–43]. The inflammatory response is linked to the racialization process in that, through racialization theory one does not reduce racial discrimination to interpersonal forms of racism but recognizes that key features of structural racism are integrated in the process of assigning political value to fictional categorizations of race[28–30]. For instance, disparities in biological markers of disease between racialized social groups are the mere physiological expressions of racism[25]. Therefore, systemic inflammation can be understood as the central mechanism linking the stress of racism to the racialized bodies of those who survive it[25]. Circulating levels of CRP have been associated with higher white matter hyperintensity[44], Alzheimer's disease[45], and all-cause dementia[46]; Elevated CRP levels are associated with chronic conditions like cardiovascular disease, adverse cognitive status, and higher risk of dementia[35–37,45,47–51]—albeit in studies of populations racialized similarly[45]. In a large cohort of individuals racialized as white of Danish descent, after adjusting for plasma lipids, health behaviors, and the genetic influence of APOE-ε4 allele carrier status, low peripheral levels of CRP were associated with a higher risk of Alzheimer's disease and all-cause dementia[45]. Other studies in European populations have found that high circulating levels of CRP were associated with a higher risk of dementia[35,36,44]. A recent study in Norway demonstrated that elevated levels of CRP may be associated with a higher risk of dementia in adults of 60 to 70.5 years of age, but this association shifted for senior adults (>70.6 years)[52]. These conflicting findings suggest that the relationship between circulating levels of CRP and dementia risk is complex and modified by factors like age[52] and cognitive domain[48]; therefore, large studies in diverse populations are warranted[16,19,20]. Some research has shown that

non-Hispanic Black women have the highest levels of CRP in comparison to non-Hispanic white women and men, and even non-Hispanic Black men[53–55]. Thus, there is reason to believe that systemic inflammation, via elevated CRP, may be important in linking the downstream effects of racialization to systemic inflammation and cognitive function[19,26].

This study examines the mediating role of systemic inflammation, and the moderating role of racialized groups on disparities in incident dementia in a large, diverse, population-based study[56,57]. During European colonization, individuals were racialized based on skin tone, perceived country or continent of origin, and/or religious affiliation[58]. This categorization created social hierarchies where a privileged racialized group (i.e., non-Hispanic white) could receive the political and socioeconomic benefits at the expense of marginalization of other groups[28,29,59–61]. Because individuals racialized as white uniquely benefit from racialization, we examined their health benefits in comparison to other racialized groups[16,59,62–66]. We expected that a lack of compounded negatives experiences of discrimination, social exclusion, and marginalization was embodied as no chronic stress response or lower CRP[26,41,67–69]. Research shows that persistent experiences of discrimination in minoritized individuals are associated with higher circulating levels of pro-inflammatory cytokines (i.e., CRP, Interleukin-6)[53,70]. This inflammatory state represents the pathway by which minoritized social groups embed the social exclusionary system in which they live. In this way systemic inflammation may be the central mechanism to understand minoritized individuals increased susceptibility to chronic conditions (i.e., cardiovascular disease, dementia, cancer) and early mortality[26,40–42,67]. Therefore, when comparing racialized groups, we are comparing forced membership in a minoritized social group (i.e., racialized non-Hispanic Black and/or Hispanic) as opposed to a more privileged one (i.e., racialized non-Hispanic white), which captures the impact of racialization on health disparities, and not fictionalized genetic or ancestry differences[71–73]. Throughout this study, we conceptualize health disparities as the result of the racialization process[21,71], therefore, we use causal mediation-interaction analysis to determine whether self-reported racialized social categories (e.g., as a proxy for exposure to racialization) is an effect modifier of the association between systemic inflammation and incident dementia, as well as to understand if systemic inflammation is a mediating pathway of disparities among racialized groups (i.e., non-Hispanic Black and/or Hispanic vs non-Hispanic white). Finally, because apolipoprotein E (APOE) is associated with lower circulating levels of CRP (our mediator)[74,75], and the carrier status of the APOE-ε4 allele confers a different cumulative risk for the development of dementia in individuals of African, Hispanic, and European ancestry[76]. We used randomized analog models, in sensitivity analyses, to test whether the APOE-ε4 allele could be better treated as a mediator-outcome confounder affected by the exposure (racialized social categories) rather than a mere confounder. Although genetic ancestry can have important effects on human health, its effects are distinct from the social construction of race[49]. However, in the Health and Retirement Study, racialized social categories are artificially paired to genetic ancestry, this feature of the data represented an opportunity to test the robustness of our main mediation analysis. In addition, we treated educational attainment as another potential mediator-outcome confounder affected by the exposure, given that in the United States educational attainment has been highly segregated, and education is an important factor associated with dementia onset[77]. We revised these sensitivity analyses in light of our main mediation models, and provide a comprehensive conceptualization for the use of CRP as a potential pathway to understand disparities in incident dementia among racialized social groups. Our results suggest that elevated levels of systemic inflammation are associated with a higher risk of dementia in US adults, and high CRP levels (≥4.73 μg/mL) explain a small proportion of the racial disparity in dementia incidence between minoritized US adults and non-Hispanic white adults.

## Methods
### Study design
The US Health and Retirement Study (HRS) is a longitudinal study of older adults in the United States[78]. To ensure representativeness of the national

demographic composition, the HRS oversamples non-Hispanic Black and Hispanic participants using a multi-stage probability design[78]. The initial cohort was formed in 1992 and interviews are conducted every 2 years. Written informed consent was obtained from all participants at data collection. Prior to each interview, participants are provided a written informed consent form. On the day of the interview, participants were read the confidentiality statement and gave oral consent by agreeing to proceed with the interview. Additionally, HRS participants provided written authorization for the collection of blood-based biomarkers, physical measures, and genetic samples. The research performed in this study was a secondary use of existing data and no additional consent or participant contact occurred. This secondary data analysis was approved as exempt and not regulated as human subjects research by the University of Michigan Institutional Review Board (HUM00128220). Survey data are publicly available (https://hrs.isr.umich.edu/data-products), and genetic data are available through dbGaP (https://dbgap.ncbi.nlm.nih.gov; phs000428.v2.p2) and the National Institute on Aging Genetics of Alzheimer's Disease Data Storage Site (https://dss.niagads.org/; NG00119).

In 2006, a random half of the participants was selected for dried blood spot and biomarker assessment and another half in 2008. For this analysis, we selected participants who provided dried blood spot samples and were cognitively normal or had cognitive impairment non-dementia (CIND) at their respective baseline (2006 or 2008). In our analytical sample, we excluded participants with prevalent dementia at baseline. Every 2 years after baseline measurements, participants underwent cognitive assessments; they were followed up either until their cognitive test results indicated dementia or until the end of a 6-year follow-up period (2012 or 2014). We explored the association between circulating CRP and 6-year incident dementia in the overall analytical sample and across three racialized groups: non-Hispanic Black, non-Hispanic white, and Hispanic participants. We used mediation-interaction analysis decomposition to explore the moderating effect of racialized categories on the association between systemic inflammation and incident dementia. We also tested whether systemic inflammation was a mediator of racial disparity in incident dementia between non-Hispanic Black and/or Hispanic participants relative to their non-Hispanic white counterparts. This study adhered to both the Strengthening the Reporting of Observational Studies in Epidemiology (STROBE) guidelines[79] and the Guideline for Reporting Mediation Analyses (AGReMA)[80].

## Measures
**Outcome**. Cognitive status was evaluated at baseline and every two years through the Telephone Interview for Cognitive Status (TICS). Cognitive test results were recorded in a continuous scale, including ten word immediate and delayed recall tests, a serial 7 s subtraction test of working memory, counting backward to assess attention and processing speed, an object naming to assess language, and recalling the date, president, and vice-president to assess orientation. All these items represent a cognitive functioning measure that ranges from 0 to 35 points, with larger values indicating better cognitive performance. We used the Langa-Weir approach to classify HRS participant's cognitive status based on a subset of the TICS assessments, specifically the immediate and delayed recall tests, serial 7 s, and backward counting. The range of scores of this subset of cognitive measures was 0–27 points[81]. According to the Langa-Weir approach, participants scoring 0–6 points were classified with dementia, 7–11 points were classified with CIND, and 12–27 points were classified as cognitively normal[81]. For the purpose of our analysis we focused solely on participants who did not have dementia at baseline (i.e., cognitively normal, or CIND) and who developed dementia over the 6-year study period. Proxy respondents were not included in our sample, as they did not provide blood spots.

## Exposure
Following current recommendations for the study of racial disparities in epidemiologic research[71], we used participants' self-reported racialized

categories as a proxy measure of exposure to the racialization process. We compared each minoritized group (non-Hispanic Black and Hispanic) to the most privileged category (non-Hispanic white, reference group)[71]. Participants racialized as Hispanic included those racialized as Hispanic white (58.6%), Hispanic Other race (37.1%), and Hispanic Black (2.8%). Although, non-Hispanic Black and Hispanic individuals are highly heterogenous groups; in the United States, they have experienced structural discrimination in the form of redlining, educational segregation, mob violence, Jim Crow and anti-immigrant laws[30,62,65,77,82–84]. These historical events have placed generations of non-Hispanic Black and Hispanic Americans behind their non-Hispanic white counterparts and are the root cause of important disparities in health and economic mobility[16,62,63,85]. To capture this minoritized status and to leverage a larger sample size to detect small statistical effects, we combined non-Hispanic Black and Hispanic participants into a minoritized category. Throughout the manuscript, when comparing jointly non-Hispanic Black and Hispanic participants to the most privileged group, we use the terminology minoritized group; otherwise, we specify which racialized groups are being compared (i.e., non-Hispanic Black vs non-Hispanic white, or Hispanic vs non-Hispanic white). In our causal diagram (Fig. 1), the arrow from the historical and institutional processes to the minoritized group membership indicates that these historical events force individual-level memberships into racially defined categories. These racialized social categories reflect hierarchies of privilege and social position rather than phenotypical, ancestral, or cultural attributes[29,58].

## Mediator
Circulating CRP was measured in blood spots using an enzyme-linked immunosorbent assay (ELISA)[86]. The CRP assay's lower limit of detection was 0.035 mg/L, the within-assay imprecision was 8.1%, and the between-assay imprecision was 11.0%. In an independent sample, this dried blood spot approach was validated against the more typical plasma sample measures ($n = 87$ paired samples, Pearson R = 0.99)[86]. Because there are no clinical thresholds for stratification of CRP in blood and dementia risk, we used the highest quartile of the distribution to denote exposure to high systemic inflammation levels. Previous studies have used the highest quartile of CRP to assess risk stratification of cerebrovascular events such as ischemic stroke, and ischemic attack[87]. In this study, we dichotomized CRP concentrations at the ≥75th percentile (highest quartile, and blood concentrations ≥4.73 µg/mL) to explore its association with incident dementia, and its mediating effect on the racial disparity. These concentrations of CRP (≥4.73 µg/mL) fall within the high stratification risk for cardiovascular events as suggested by the Center for Disease Control and Prevention (CDC) and the American Heart Association (AHA)[88].

## Covariates
We included potential confounders of the association between our exposure of interest, racialized groups, our mediator CRP, and incident dementia, with all confounders measured at baseline (Fig. 1). Sociodemographic confounders were self-reported and included age (continuous, in years, calculated from birth date and interview date), sex (female or male), and education (more than college, college or some college, high school or less). Behavioral confounders included smoking status (current, former, never), alcohol consumption (reported as number of drinks a day when drinks, continuous), and self-reported body mass index (calculated as weight kilograms divided by height in meters squared, continuous). Number of chronic health conditions included high blood pressure (yes or no), diabetes (yes or no), cancer (yes or no), lung disease (yes or no), heart disease (yes or no), stroke (yes or no), psychiatric problems (yes or no), and arthritis (yes or no), and was operationalized in our models as a continuous variable (0–8 conditions). Genetic information on APOE-ε4 allele carrier status (at least one copy or no copy) was obtained from phased genetic data imputed to the worldwide 1000 Genomes Project reference panel. Genotyping and imputation information on the Health and Retirement

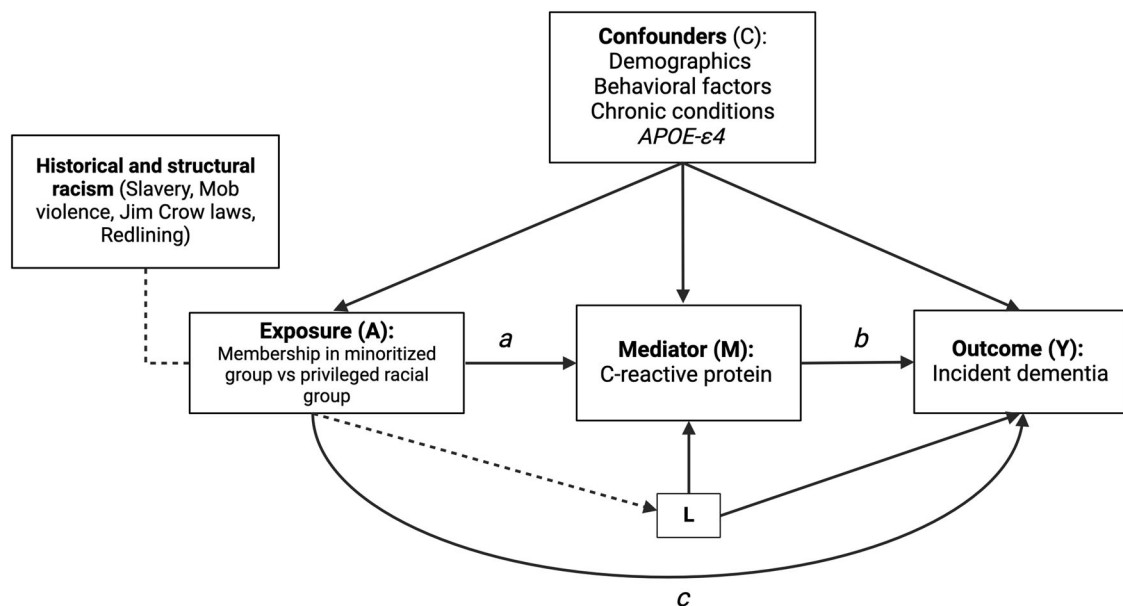

**Fig. 1 | Directed acyclic graph illustrating the relationship between racialized social groups, systemic inflammation, and 6-year incident dementia in the Health and Retirement Study.** Caption: Exposure (A) represents membership in a minoritized or racialized group vs a privileged group (i.e., non-Hispanic Black and/ or Hispanic participants vs non-Hispanic white participants). Racialized group membership stems from historical and structural processes related to racism and this forced membership status is directly associated with incident dementia, as denoted in arrow c, and through systemic inflammation (Mediator M) as denoted by arrows (a) and (b). The association between systemic inflammation and incident dementia is denoted by arrow (b). However, the association between systemic inflammation and incident dementia can be modified by membership in minoritized racial status, as this model allows for exposure-mediator interaction. The set of confounders (C) account for exposure (A) - outcome (Y), exposure (A) - mediator (M), and mediator (M) - outcome (Y) confounders. This model also assumes that there are no mediator-outcome confounders (L) affected by the exposure (A).

Study is available elsewhere[89]. We also included baseline survey wave (2006 or 2008) as a covariate to account for unmeasured differences across waves.

## Statistical analysis

Our analytical sample included participants with complete information for all our covariates of interest and those who developed dementia over the 6-year period from either a cognitively normal status or CIND. We examined the distributions of all baseline covariates by each of the self-reported racialized categories, quartiles of the CRP distribution, and incident dementia using bivariate statistical tests, as appropriate. We used kernel density plots to explore the distribution of CRP by the three racialized groups, as well as stratified by *APOE-ε4* allele carrier status and racialized categories. Additionally, we explored the distribution of CRP concentrations by racialized groups and self-reported sex categories using frequency statistics. We dichotomized CRP at the 75th percentile of the study sample distribution and categorized those with levels above or equal to the 75th percentile (≥4.73 μg/mL) as high, and those less than the 75th percentile (<4.73 μg/mL) as low. Because our primary endpoint of interest was incident dementia, we excluded participants with prevalent dementia at baseline, and those with incomplete information on covariates of interest (Supplemental Fig. 1).

In the overall sample and stratified by either minoritized status or racialized groups, we employed multivariable-adjusted Poisson regression models with a logarithmic link function and time-to-dementia as an offset variable, to estimate incident rate ratios of dementia between participants with high CRP (≥75th percentile) versus low CRP. In order to understand how the magnitude of the association between our mediator (CRP) and our outcome (incident dementia) changed with different sets of confounders, we fitted four sequential regression models: an unadjusted model, a demographic model (adjusted for age, sex, education, *APOE-ε4* allele status, and survey wave), a behavioral model (demographic adjusted model, and smoking status, alcohol consumption, and body mass index), and a chronic condition model (behavioral adjusted model, and chronic conditions).

We employed logistic regression analysis to estimate the association between each racialized group (each of non-Hispanic Black and Hispanic, versus non-Hispanic white) and minoritized group (non-Hispanic Black or Hispanic, versus non-Hispanic white) and the odds of high CRP levels (≥75th percentile), adjusted for the same confounders as described above. We performed a four-way mediation-interaction decomposition analysis to evaluate whether CRP mediated disparities among racialized groups in incident dementia using the CMAverse R studio package, accounting for any interaction effect between minoritized group status and CRP[90]. This interaction effect allowed us to capture whether belonging to a minoritized group differentially affected the strength of the association between systemic inflammation and incident dementia. Decomposition estimates were obtained using the *cmest* function of the CMAverse package, and employing the regression-based approach and direct counterfactual imputation for estimation[90]. The 95% confidence intervals of our estimates were calculated using the percentile bootstrapping inference method, we performed 1000 bootstraps in each procedure and set a random seed for reproducibility purposes. Analyses were conducted using R statistical software (version 3.6.2) and Stata (v17). A second analyst performed complete code review. Code to produce these analyses is available (https://github.com/bakulskilab/Racialization_CRP_Dementia)[91].

## Sensitivity analysis

Mediation analysis assumes that if the adjustment set of covariates is sufficient to control for all exposure-outcome, mediator-outcome, and exposure-mediator confounders, then natural or pure indirect effects are identifiable[92]. However, an additional assumption to identify indirect effects is needed: no mediator-outcome confounder should be affected by the exposure[92]. Our directed acyclic graph (Fig. 1) illustrates a potential scenario in which a variable (L) acts as a mediator-outcome confounder affected by the exposure. In the HRS, self-reported racialized categories are paired to geographic genetic ancestry groups by study design[89]. Because non-Hispanic Black participants of African ancestry are more likely to be carriers of the

**Table 1 | Distribution of baseline sample characteristics by dementia status after 6 years of follow-up, United States Health and Retirement Study, 2006 and 2008**

| Characteristic | Overall Sample N = 6908[a] | Incident dementia N = 795[a] | Cognitively Normal or CIND N = 6113[a] | p value[b] |
|---|---|---|---|---|
| Baseline CRP (µg/mL) | 4.37 (8.00) | 5.13 (11.28) | 4.27 (7.46) | 0.004 |
| Age (years) | 67.11 (9.91) | 75.64 (9.62) | 66.00 (9.40) | <0.001 |
| Race | | | | <0.001 |
| Non-Hispanic Black | 813 (12%) | 171 (22%) | 642 (11%) | |
| Hispanic | 633 (9.2%) | 103 (13%) | 530 (8.7%) | |
| Non-Hispanic white | 5462 (79%) | 521 (66%) | 4941 (81%) | |
| Sex | | | | 0.473 |
| Female | 4234 (61%) | 478 (60%) | 3756 (61%) | |
| Male | 2674 (39%) | 317 (40%) | 2357 (39%) | |
| Educational Category | | | | <0.001 |
| > College | 676 (9.8%) | 25 (3.1%) | 651 (11%) | |
| College/Some | 1297 (19%) | 68 (8.6%) | 1229 (20%) | |
| HS or < | 4935 (71%) | 702 (88%) | 4233 (69%) | |
| Alcohol Use (# drinks/day when drinks) | 0.71 (1.30) | 0.40 (1.07) | 0.75 (1.32) | <0.001 |
| Smoking Status | | | | 0.430 |
| Current | 851 (12%) | 104 (13%) | 747 (12%) | |
| Former | 2966 (43%) | 352 (44%) | 2614 (43%) | |
| Never | 3091 (45%) | 339 (43%) | 2752 (45%) | |
| Body Mass Index (kg/m²) | 28.50 (5.79) | 27.17 (5.38) | 28.68 (5.82) | <0.001 |
| Chronic Conditions | 1.88 (1.36) | 2.47 (1.51) | 1.80 (1.32) | <0.001 |
| *APOE-ε4* | | | | <0.001 |
| At least 1 copy | 1859 (27%) | 258 (32%) | 1601 (26%) | |
| No copy | 5049 (73%) | 537 (68%) | 4512 (74%) | |
| Time (years) | | | | <0.001 |
| 2 | 302 (4.4%) | 302 (38%) | 0 (0%) | |
| 4 | 269 (3.9%) | 269 (34%) | 0 (0%) | |
| 6 | 6337 (92%) | 224 (28%) | 6113 (100%) | |
| Wave | | | | <0.001 |
| 2006 | 3485 (50%) | 339 (43%) | 3146 (51%) | |
| 2008 | 3423 (50%) | 456 (57%) | 2967 (49%) | |

*CRP* C-reactive protein, *APOE-ε4* apolipoprotein E ε4 allele carrier status, *CIND* cognitive impairment non-dementia

[a]Mean (SD); *n* (%)

[b]One-way ANOVA; Pearson's Chi-squared test

*APOE-ε4* allele (variable L, Fig. 1), and this allele is associated with both circulating levels of CRP (mediator) and dementia (outcome), we employed randomized analog models to test the robustness of our mediation analysis findings[92]. In these models, decomposition estimates were obtained employing the g-formula approach and direct counterfactual imputation for estimation. The 95% confidence intervals of these randomized analog model estimates were calculated using the percentile bootstrapping inference method, we performed 1000 bootstraps in each procedure and set a random seed for reproducibility purposes. Furthermore, educational attainment has historically been racialized and segregated in the United States, and research has shown that this socioeconomic health determinant is associated with

both systemic inflammation and adverse cognitive outcomes[77,93–95]. In an additional sensitivity randomized analog model, we treated both educational attainment and *APOE-ε4* allele as mediator-outcome confounders affected by the exposure. Additionally, we conducted different sensitivity models, we estimated incidence rate ratios (IRR) for elevated CRP but with cognitive impairment as the outcome, meaning CIND cases were included with dementia cases and compared to cognitively normal participants. Because of the well-described issues on dichotomizing continuous exposures in health science research[96,97], we also estimated incident rate ratios of dementia using the standardized natural logarithmic transformation of CRP as the primary predictor; and re-calculated the regression-based and g-estimation mediation-interaction decomposition models setting the mediator levels at zero (or the mean) and to be evaluated at one (or one SD above the mean) for the estimation of the controlled direct effect. Finally, because our primary interest is the identification of the mediated effect of CRP on disparities, we calculated mediational E-values to examine the extent to which an unmeasured confounder could explain away our observed mediational effect[98]. Mediational E-values were calculated for the total natural indirect effect (proportion mediated) for both regression-based and randomized analog estimates.

## Results

### Sample characteristics

Our analytic sample size included 6,908 participants (Supplemental Fig. 1). On average, participants were 67.1 years of age, 61% were female, 12% were non-Hispanic Black, 9.2% were Hispanic, 71% completed high school education or less, and had average CRP concentrations of 4.37 µg/mL (Table 1). Excluded participants had higher circulating CRP, were older, more likely to be male, non-Hispanic Black or Hispanic, and had completed high school education or less (Supplemental Table 1). Cumulative dementia incidence over the 6-year follow-up was 21% for non-Hispanic Black participants, 16% for Hispanic participants, and 9.5% for non-Hispanic white participants (Supplemental Table 2). The highest mean CRP levels were observed in non-Hispanic Black participants (6.5 µg/mL) followed by Hispanic participants (4.5 µg/mL) and non-Hispanic white participants (4.1 µg/mL) (Supplemental Table 2 and Fig. 2). Non-Hispanic Black female participants had the highest average concentrations of CRP levels (6.8µg/mL), whereas non-Hispanic white male participants had the lowest (3.4 µg/mL). This hierarchy of systemic inflammation followed a racial and sex gradient, a phenomenon we call the racial hierarchy of inflammation (Supplemental Table 3 and Supplemental Fig. 2).

### Associations between C-reactive protein and incident dementia in the overall analytic sample, by minoritized and racialized groups

On average, participants with incident dementia had higher levels of CRP (5.1 µg/mL), compared to participants with normal cognition (4.3 µg/mL) (Table 1). However, the proportion of incident dementia cases across quartiles of the CRP distribution did not substantially differ. For example, among those with CRP levels ≥4.73 µg/mL (≥75th percentile), the 6-year cumulative dementia incidence was 13%, and among those with CRP levels <0.98 µg/mL (<25th percentile) was 12% (Supplemental Table 4). Participants in the 75th percentile of the CRP distribution were, on average, of slightly younger age, female, non-Hispanic white, and completed a high school education or less compared to those below the 25th percentile. Additionally, these participants were more likely to be current smokers, had fewer drinks per day when they drink, had a larger body mass index on average, were more likely to have a higher average number of health conditions, and were less likely to be carriers of the *APOE-ε4* allele than those in the lowest quartile of the distribution (<25th percentile) (Supplemental Table 4).

In our overall sample, the fully adjusted model showed that among those exposed to high inflammation levels, the 6-year risk of incident dementia was 1.23 (95% CI: 1.05, 1.44) times higher than in those with low inflammation levels (Table 2). Sequential adjustment suggested that the

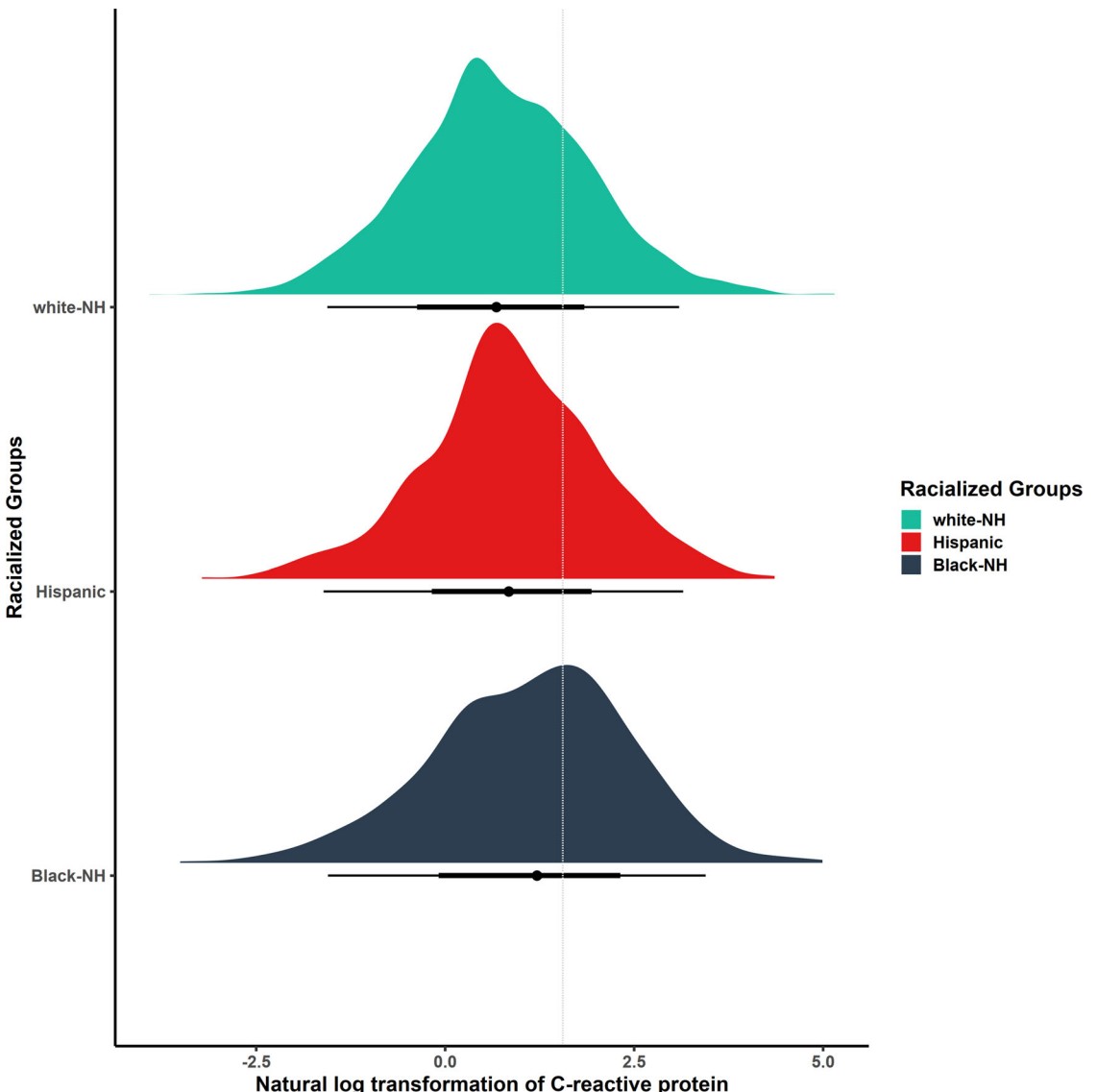

**Fig. 2 | Density plot visualizing the distribution of the natural logarithmic transformation of C-reactive protein (CRP) by racialized social groups in the Health and Retirement Study.** Caption: Dotted line denotes the cutoff point for elevated levels of CRP at the 75th percentile (≥4.73 μg/mL, *n* = 8,320). white-NH non-Hispanic white (*n* = 6602), Black-NH non-Hispanic Black (*n* = 971), Hispanic (*n* = 747).

strength of the association between high CRP levels and dementia risk increased after conditioning on potential confounders. The association between high CRP levels and incident dementia differed across minoritized and racialized groups. For example, among minoritized participants, high CRP was associated with 1.26 (95% CI: 0.98, 1.62) times higher risk of incident dementia than low CRP, although this finding was not statistically significant. Similarly, the risk of 6-year incident dementia for non-Hispanic white participants with high CRP was 1.19 (95% CI: 0.98, 1.45) times higher than those with low CRP, but this finding was not statistically significant. However, among Hispanic participants, high CRP was associated with 1.85 (95% CI: 1.27, 2.70) times higher risk of dementia than low CRP. Among non-Hispanic Black participants, the association between CRP and incident dementia was null (IRR: 1.00; 95%CI: 0.72, 1.37) (Table 2).

**Minoritized and racialized disparities in high circulating levels of C-reactive protein**
In multivariable-adjusted models, we found that minoritized participants had 1.37 (95% CI: 1.19, 1.58) times higher odds of elevated CRP compared to non-Hispanic white participants. When each racialized group was analyzed separately, we found that non-Hispanic Black participants had 1.70 (95% CI:

1.43, 2.02) times higher odds of elevated CRP than their non-Hispanic white counterparts. However, this association was null for Hispanic participants (OR: 1.00; 95%CI: 0.81, 1.23); we found that differences in high levels of inflammation between Hispanic participants and their non-Hispanic white counterparts were accounted for when demographic confounders were included in the model (Supplemental Table 5).

**Four-way mediation-interaction decomposition to assess C-reactive protein as a mediator of the racialized disparities in incident dementia**
In our fully adjusted regression-based mediation models, the decomposition analysis comparing minoritized versus non-minoritized groups showed that the mediating effect of CRP accounted for 3% (95% CI: 0%, 6%) of the disparity in incident dementia, while the interaction effect between minoritized group status and elevated CRP accounted for 14% (95% CI: 1%, 27%) of the disparity (Table 3 and Supplemental Fig. 3). When decomposing the non-Hispanic Black vs non-Hispanic white disparity, we found that the mediating effect of CRP accounted for 2% (95% CI: −3%, 8%) of the disparity, and the portion attributable to the interaction accounted for 4% (95% CI: −11%, 21%), while neither of these estimates was statistically

**Table 2 | Incidence rate ratios from Poisson regression analysis, estimates represent the association between elevated levels of C-reactive protein (CRP) (≥4.73 µg/mL) and 6-year incident dementia in the United States Health and Retirement Study**

| | Incident Dementia | | | | | | | | | |
|---|---|---|---|---|---|---|---|---|---|---|
| | Overall *N* = 6908 | | Minoritized (non-Hispanic Black & Hispanic) *N* = 1446 | | Non-Hispanic Black *N* = 813 | | Hispanic *N* = 633 | | Non-Hispanic white *N* = 5462 | |
| **Models** | | | | | | | | | | |
| **Unadjusted** | IRR[a] | 95% CI[a] | IRR[a] | 95% CI[a] | IRR[a] | 95% CI[a] | IRR[a] | 95% CI[a] | IRR[a] | 95% CI[a] |
| <75th | 1 | - | 1 | - | 1 | - | 1 | - | 1 | - |
| ≥75th (≥4.73 µg/mL) | 1.13 | [0.97,1.32] | 1.11 | [0.87,1.41] | 0.83 | [0.61,1.13] | 1.67** | [1.13,2.46] | 1.00 | [0.82,1.23] |
| **Demographic[b]** | | | | | | | | | | |
| <75th | 1 | - | 1 | - | 1 | - | 1 | - | 1 | - |
| ≥75th (≥4.73 µg/mL) | 1.21* | [1.04,1.41] | 1.26 | [0.99,1.60] | 0.99 | [0.72,1.34] | 1.80** | [1.23,2.65] | 1.19 | [0.98,1.45] |
| **Risk Factors[c]** | | | | | | | | | | |
| <75th | 1 | - | 1 | - | 1 | - | 1 | - | 1 | - |
| ≥75th (≥4.73 µg/mL) | 1.25** | [1.07,1.46] | 1.27 | [0.99,1.63] | 0.99 | [0.72,1.37] | 1.89** | [1.29,2.77] | 1.23* | [1.01,1.50] |
| **Chronic Conditions[d]** | | | | | | | | | | |
| <75th | 1 | - | 1 | - | 1 | - | 1 | - | 1 | - |
| ≥75th (≥4.73 µg/mL) | 1.23** | [1.05,1.44] | 1.26 | [0.98,1.62] | 1.00 | [0.72,1.37] | 1.85** | [1.27,2.70] | 1.19 | [0.98,1.45] |

[a]IRR: incidence rate ratio, CI: confidence interval in brackets,

[b]Demographic model: adjusted for age, sex, education categories, *APOE-ε4* allele status, and wave. Note: the demographic model in the overall sample (*N* = 6908) additionally adjust for racialized social groups

[c]Risk factors model: adjusted for age, sex, education categories, *APOE-ε4* allele status, wave, smoking status, alcohol consumption, body mass index.

[d]Chronic conditions model: adjusted for age, sex, education categories, *APOE-ε4* allele status, wave, smoking status, alcohol consumption, body mass index, and chronic conditions

*p < 0.05, **p < 0.01, ***p < 0.001

Models are stratified by racialized social groups and minoritized status.

significant (Table 3 and Supplemental Fig. 3). The Hispanic vs non-Hispanic white decomposition showed that the mediating effect of CRP was virtually zero. However, the portion attributable to the interaction accounted for 28% (95% CI: 8%, 51%) of the disparity (Table 3 and Supplemental Fig. 3).

**Sensitivity analysis**

In this analysis, *APOE-ε4* may be a mediator-outcome confounder affected by our exposure of interest through racialized status being paired to geographic ancestry (**variable L**, Fig. 1 and Supplemental Fig. 4). Because of this potential violation of causal mediation analysis, we conducted a randomized analog mediation model to test the robustness of our regression-based mediation estimates. We found that when comparing the minoritized group to the non-Hispanic white group, the mediating effect of CRP on incident dementia accounted for 4% (95% CI: 0%, 6%) of the disparity, and the proportion due to interaction accounted for 15% (95% CI: 1%, 29%) (Supplemental Table 6 and Supplemental Fig. 3). When decomposing the non-Hispanic Black vs non-Hispanic white disparity, we found that the mediating effect of CRP accounted for 2% (95% CI: −3%, 7%) of the disparity, and the portion attributable to the interaction accounted for 5% (95% CI: −12%, 23%), but these estimates were not statistically significant (Supplemental Table 6 and Supplemental Fig. 3). The Hispanic vs non-Hispanic white decomposition showed that the mediating effect of CRP accounted for 1% (95% CI: −8%, 6%), and the portion attributable to the interaction accounted for 30% (95% CI: 9%, 57%) of the disparity (Supplemental Table 6 and Supplemental Fig. 3). These results were similar to those obtained from our regression-based estimates, suggesting that our findings were robust to a potential violation of mediation analysis. Moreover, our mediational E-value suggested that an unmeasured confounder associated with both high CRP and incident dementia with approximate rate ratios of 1.16-fold could completely explain away the observed indirect effect of the minoritized group vs. non-Hispanic white disparity, but a weaker confounder could not (Supplemental Table 7A). Further, an unmeasured confounder associated with both high CRP and incident dementia with approximate rate ratios of 1.06-fold could shift the mediated

proportion confidence interval to the null, but a weaker confounder could not. Mediational E-values for the randomized analog models were of slightly similar magnitude (Supplemental Table 7B).

Moreover, because educational attainment has historically been racialized and segregated in the United States, and segregated schooling might be associated with both dementia, and C-reactive protein; we similarly conducted a randomized analog mediation model to test the robustness of our mediated and interaction effects. This sensitivity model assumes that more than one confounder (i.e., *APOE-ε4* and educational attainment) was affected by the exposure (racialization process). We found that when comparing the minoritized group to the non-Hispanic white group, the mediating effect of CRP on incident dementia accounted for 3% (95% CI: 0%, 6%) of the disparity, and the proportion due to interaction accounted for 15% (95% CI: 2%, 28%) (Supplemental Table 8). We obtained similar estimates for the decomposition effects of the non-Hispanic Black vs non-Hispanic white disparity, and the Hispanic vs non-Hispanic white disparity than in the randomized analog models that only included *APOE-ε4* as a sole mediator-outcome confounder affected by the exposure (Supplemental Tables 8, 6). In a sensitivity model exploring the association between high CRP protein levels and 6-year cognitive impairment (i.e., dementia and cognitive impairment non-dementia cases vs cognitively normal participants), we did not observe statistically significant associations in the overall model or in stratified models by racialized social groups (Supplemental Table 9), which may indicate that CRP may have a more important effect in differentiating dementia cases from cognitive impairment non-dementia cases. Lastly, in a fully adjusted model, we found that 1 standard deviation above the mean log-transformed CRP was associated with 1.06 (95%CI: 0.99, 1.14) times greater risk of dementia in the overall sample. This estimate was of slightly larger magnitude for non-Hispanic white participants (IRR = 1.07; 95%CI: 0.98, 1.16) in comparison to their non-Hispanic Black (IRR = 1.03; 95%CI: 0.89, 1.19) and Hispanic (IRR = 1.03; CI: 0.89, 1.16) counterparts (Supplemental Table 10). Nonetheless, none of these estimates achieved the statistically significant threshold of 5%. When decomposing the minoritized versus non-Hispanic white disparity, the non-Hispanic Black versus the non-Hispanic white disparity, and the Hispanic versus the

**Table 3 | Four-way mediation analysis decomposition for racialized disparities in incident dementia using elevated levels of C-reactive protein (CRP ≥4.73 µg/mL) as a mediator**

| Mediator (CRP) | Incident dementia* | | | | | | | | | | | |
| --- | --- | --- | --- | --- | --- | --- | --- | --- | --- | --- | --- | --- |
| | Minoritized racial group vs non-Hispanic white N = 6908 | | | | non-Hispanic Black vs non-Hispanic white N = 6275 | | | | Hispanic vs non-Hispanic white N = 6095 | | | |
| | Estimate | [95%CI] | [95%CI] | p value | Estimate | [95%CI] | [95%CI] | p value | Estimate | [95%CI] | [95%CI] | p value |
| **Excess risk** | | | | | | | | | | | | |
| RERI Controlled Direct Effect | 1.26 | 0.91 | 1.67 | <0.00 | 1.60 | 1.15 | 2.21 | <0.00 | 0.94 | 0.48 | 1.51 | <0.00 |
| RERI Interaction Reference | 0.16 | 0.01 | 0.32 | 0.04 | 0.06 | −0.14 | 0.26 | 0.66 | 0.39 | 0.10 | 0.71 | 0.01 |
| RERI Interaction Mediation | 0.04 | 0.00 | 0.08 | 0.04 | 0.02 | −0.06 | 0.11 | 0.66 | −0.02 | −0.09 | 0.06 | 0.82 |
| RERI Pure Indirect Effect | 0.01 | 0.00 | 0.02 | 0.23 | 0.01 | 0.00 | 0.04 | 0.13 | 0.00 | −0.01 | 0.01 | 0.85 |
| **% Attributable** | | | | | | | | | | | | |
| % Controlled Direct Effect | 0.86 | 0.73 | 0.99 | <0.00 | 0.95 | 0.78 | 1.10 | <0.00 | 0.72 | 0.49 | 0.92 | <0.00 |
| % Interaction Reference | 0.11 | 0.00 | 0.22 | 0.04 | 0.03 | −0.08 | 0.15 | 0.66 | 0.30 | 0.07 | 0.52 | 0.01 |
| % Interaction Mediation | 0.03 | 0.00 | 0.06 | 0.04 | 0.01 | −0.04 | 0.07 | 0.66 | −0.02 | −0.07 | 0.05 | 0.82 |
| % Pure Indirect Effect | 0.00 | 0.00 | 0.01 | 0.23 | 0.01 | 0.00 | 0.02 | 0.13 | 0.00 | −0.01 | 0.01 | 0.85 |
| Percent Mediated | 0.03 | 0.00 | 0.06 | 0.01 | 0.02 | −0.03 | 0.08 | 0.42 | −0.02 | −0.08 | 0.05 | 0.82 |
| Percent due to Interaction | 0.14 | 0.01 | 0.27 | 0.04 | 0.04 | −0.11 | 0.21 | 0.66 | 0.28 | 0.08 | 0.51 | 0.01 |
| Percent Eliminated | 0.14 | 0.01 | 0.27 | 0.03 | 0.05 | −0.10 | 0.22 | 0.55 | 0.28 | 0.08 | 0.51 | 0.01 |

*Outcome 6-year incident dementia, model adjusting for age, sex, education categories, APOE-ε4 allele status, wave, smoking status, alcohol consumption, body mass index, and chronic conditions. Minoritized racial group: (non-Hispanic Black and Hispanic participants)
Models are stratified by minoritized status and racialized social groups in a sample of United States adults in the Health and Retirement Study

non-Hispanic white disparity using the natural log-transformed CRP variable in regression-based (Supplemental Table 11) and randomized analog models (Supplemental Table 12), we did not find statistically significant evidence of interaction or mediation, suggesting that the underlying interaction and mediated effects between racialized social categories and systemic inflammation may be dependent on a particular threshold of systemic inflammation.

## Discussion

Disparities in dementia among racialized groups are the result of multiple expressions of racism, and unveiling the biological mechanisms implicated in the production of these disparities is crucial for understanding how racism is embodied[99]. In a nationally representative sample of older adults in the United States, we observed a 23% greater risk of incident dementia among those with high versus low CRP, and this association was stronger among Hispanic and non-Hispanic white participants than among non-Hispanic Black participants. We found that 14% of the observed disparity in incident dementia was accounted for by the interaction between minoritized group membership and elevated CRP, and 3% of the disparity was mediated by high CRP. A stronger interaction effect was apparent in the Hispanic versus non-Hispanic white decomposition, where we found that 28% of the disparity was attributable to the interaction effect between Hispanic group membership and high CRP. When decomposing the non-Hispanic Black versus non-Hispanic white disparity, we observed that 4% was attributable to the interaction effect between non-Hispanic Black membership and high CRP, but this effect was not statistically significant. Altogether, these results indicate that systemic inflammation is associated with dementia risk, and the effect of high CRP on dementia is moderated by minoritized group status. When individuals are racialized as non-Hispanic Black and/or Hispanic, the effect of CRP on incident dementia risk is greater than expected had these individuals been racialized (and treated) as non-Hispanic white[17].

Our findings fit with previous epidemiological studies describing differences in CRP levels across racialized groups[37,48,53,100]. We found that non-Hispanic Black participants had higher circulating CRP than non-Hispanic white participants after adjusting for a wide range of covariates. These findings are consistent with those from another recent HRS analysis[24]. Additionally, our results extend prior research linking systemic inflammation and dementia risk in large population-based studies. For instance, in a nested case-control study of Japanese American men ($N = 1050$), CRP levels of >1.0 mg/L (vs. <0.34 mg/L) were associated with 2.8 times greater odds of all dementia subtypes after adjusting for sociodemographic conditions, behavioral factors, and *APOE-ε4* carrier status[37]. In a separate sample of community-dwelling older adults with a large number of non-Hispanic Black ($N = 1255$) and non-Hispanic white ($N = 1776$) participants, individuals in the highest tertile of CRP (2.5–85.2 mg/L) had 1.41 greater odds of cognitive decline than participants in the lowest tertile (0.2–1.2 mg/L), although no interaction effect between racialized group and inflammation was observed[100]. Similarly, in a racially diverse sample of the Reasons for Geographic And Racial Differences in Stroke (REGARDS) cohort, average CRP levels were higher among Black participants (2.8 mg/L; $N = 7974$) in comparison to their white counterparts (1.8 mg/L; $N = 13,808$)[48]. Using race-specific CRP cutoffs at the 90th percentile, participants with baseline CRP at or above the 90th percentile experienced a faster decline in memory and verbal fluency trajectories than those with CRP levels below the 90th percentile[48]. Again, in this study, researchers concluded that no interaction between racialized groups and inflammation on cognition was present. Altogether, these prior studies suggest that elevated systemic inflammation is associated with adverse cognitive outcomes in older adults, and this effect was not modified by racialized groups. We expanded on these previous studies by incorporating a measure of additive interaction in our mediation models[101] to test if the effect of high CRP levels on incident dementia was modified by the racialization process. This approach aligns with current epidemiological frameworks suggesting that effect modification is scale dependent, and the additive scale is better suited to test for interaction effects[102], and with more recent developments in mediation analysis that

unify mediation and additive interaction into a unique framework[101]. For instance, this innovative methodology allowed us to examine if the racialization process (implied in racialized group categories) modified the association between systemic inflammation and incident dementia, while simultaneously exploring whether systemic inflammation was a mediating pathway of the observed disparities.

Our results support the hypothesis that systemic inflammation is a plausible biological pathway implicated in the production of disparities in incident dementia. We found evidence that 3% of the disparity between the minoritized and privileged groups was attributable to the mediated CRP pathway, and another 14% was attributable to the moderated pathway. The slight mediation effect was expected since disparities between these groups emerge from structural forces acting differentially on groups rather than physiological processes that might be different among groups. These structural forces operate tacitly under the controlled direct effect, which represents a large proportion (88%) of the observed minoritized disparity. Systemic inflammation, and likely other biological responses, represent plausible mechanisms through which racism operates. In this study, we solely focused attention on a single biomarker of inflammation, but current research suggest that multiple inflammatory cytokines are related to Alzheimer's risk[103,104]; and population-based studies that incorporate multiple inflammatory mediators as pathways to understand racialized disparities in dementia risk could be better suited to detect larger mediated effects. The interplay between different racialized experiences and treatment with multiple cytokines measures deserves further attention. Our results suggest that even though non-Hispanic Black participants had higher levels of CRP than their non-Hispanic white counterparts, their risk of dementia was not statistically significant when comparing those with high CRP to those with low CRP. These results indicate that in future work, studies should incorporate multiple systemic inflammatory biomarkers across diverse racialized groups to characterize the etiological role of peripheral immunity on neurodegenerative diseases. The moderated pathway reflects the extent to which minoritized group status affects the association between CRP and incident dementia, which is greater than expected for individuals minoritized and racialized as non-Hispanic Black and/or Hispanic had these individuals been racialized and treated as non-Hispanic white. In other words, had all groups been treated comparably as non-Hispanic white individuals, disparities in incident dementia would be reduced.

Though there is a complex reality when examining CRP as a biological pathway. We found that high levels of systemic inflammation were associated with incident dementia in the overall sample, in the minoritized group, the Hispanic group, and the non-Hispanic white group. Although participants racialized as non-Hispanic Black had the highest levels of systemic inflammation, elevated CRP protein was not associated with incident dementia. An explanation for this finding may be that adults racialized as non-Hispanic Black exhibit high systemic inflammation levels as a result of high-effort coping against the stress of racism but this translated only superficially into changes in their cognitive test scores, a syndrome known in the literature as John Henryism[105–107]. Also, the majority of incident dementia cases occurred in participants racialized as non-Hispanic white ($n = 521$ or 65.5%), and to a lesser extent in participants racialized as non-Hispanic Black ($n = 171$ or 21%) and Hispanic ($n = 103$, 13%), the lower number of events in minoritized participants suggest that these statistical power issues may be a limitation. Moreover, in stratified mediation models, we did not observe statistically significant mediated or moderated effects when comparing the non-Hispanic Black versus non-Hispanic white disparity than when comparing the minoritized disparity. We attribute these findings to the null association between CRP and incident dementia among non-Hispanic Black participants, for which we have other possible explanations. We hypothesize that the higher levels of CRP found in non-Hispanic Black participants are characteristic of a chronic stress response that results from persistent experiences with structural racism[21,41,68]. Therefore, chronic systemic inflammation may predispose Black participants to other competing events such as diabetes, cardiovascular disease, stroke, and premature death[51,108,109]; which in turn may affect Black

participants' likelihood of retention during the study period. Although our models controlled for confounding bias by these potential competing events, we did not account for selection bias issues in our analysis, and future research should inform how differential loss to follow-up affects the relationship between systemic inflammation and dementia in Black participants. Additionally, we attribute the lack of mediation effect in the Hispanic vs non-Hispanic white disparity to the fact that differences in high CRP between these two groups were accounted for by individual-level confounders. However, we were able to detect important moderating effects for this disparity. The substantial heterogeneity in the relationship between high CRP and dementia risk across the distinct racialized experiences and treatment of minoritized social groups shows that Hispanic participants are more susceptible to the effect of high CRP levels on cognitive health than non-Hispanic Black participants. This suggests that unique racialized processes link biological pathways to health outcomes. Studies should further explore how diverse racialized experiences (i.e., immigration, segregation, unemployment, underemployment) influence inflammatory-related pathways and their relation to cognitive health in minoritized populations. It is noteworthy to mention that when comparing the minoritized status of non-Hispanic Black and Hispanic participants together to the most privileged social position of non-Hispanic white participants, the effect of systemic inflammation on dementia risk was moderated by the participants' position on this binary spectrum, demonstrating that on average minoritized group membership influence inflammatory pathways and brain health. Although we found consistent evidence that CRP was an important mediator of disparities in the minoritized racialized group, our research was limited to a baseline measurement and a unique biomarker. Another important limitation is that our models did not adjust for childhood socioeconomic status, and research suggests that this may be an important confounder between inflammation and dementia[110,111]. However, in sensitivity models using *APOE-ε4* and educational attainment as mediator-outcome confounders affected by the exposure, we did not observe significant changes in the magnitude of the mediating effect of systemic inflammation, or the moderating effect of the racialization process on the racial disparity. Our sensitivity models comparing minoritized status to the most privileged group yielded statistically significant results for the proportion due to mediation and interaction, these results were of similar magnitude to the main mediation models in which a potential violation of mediation analysis was ignored. However, it is noteworthy that important health determinants such as educational attainment, neighborhood characteristics, and childhood socioeconomic status are, by a large degree, driven by historical and structural processes that stem from racism[62,82]. It is difficult to identify the indirect effect of systemic inflammation on the racial disparity in incident dementia without relaying in the strong assumptions drawn in our causal diagram (Fig. 1), and the temporal relationships between confounders and mediator. Structural racism, through its multiple expressions, is the root cause of economic disparities and physiological disruptions that may affect racialized individuals' susceptibility to disease[77,82,85]. In this case, educational attainment, and other social health determinants (i.e., childhood socioeconomic status) can be understood as mediator-outcome confounders affected by the exposure. Some of these health determinants are not included in our DAG and may be operating under the controlled direct effect of the racialization process. Racist policies and historical events such as redlining, mob violence, Jim Crow, and anti-immigration laws have placed individuals racialized as non-Hispanic Black, Hispanic, and Indigenous at generational economic disadvantage and political underrepresentation. We argue that the cumulative effect of these disadvantages may have negative repercussions for the stress response with downstream consequences for cognitive aging in minoritized racialized groups; instead, non-Hispanic white participants have benefited from their racialized privileged status. Our research framework is innovative, not in that we accounted for every possible confounding variable to identify the mediating effect of systemic inflammation on racial disparity. But, in that we integrated a downstream biological determinant to understand the physiological underpinnings of the racialization process (i.e., the process of racializing individuals and differentially treating them across

multiple domains of the social life)[17,71]. Future research should expand on integrating structural measures of racism with biomarkers of disease to better capture the multiple biological expressions of racism, and its deleterious effects in human physiology[112]. Additionally, mediation analysis research should incorporate multiple biomarkers of systemic inflammation, and time-ordered confounders to better understand how early life-exposure to racism may influence systemic inflammation, and the cognitive trajectories of older adults. And, because participants at the intersection of multiple marginalized identities (i.e., non-Hispanic Black women and Hispanic women) exhibited higher levels of systemic inflammation, future work should characterize the role of racism and sexism in inflammation trajectories and dementia risk. Lastly, during the preclinical phase of Alzheimer's disease (20 years) there are changes in cerebrospinal fluid concentrations of Amyloid-β42 and other inflammatory biomarkers that are predictive of disease onset[113,114]. In large observational studies like the Health and Retirement Study, participants are routinely screened for changes in their cognitive function; clinical impairment debuts with changes in cognitive test scores starting approximately 6 years before symptoms onset[113]. We used a 6-year follow-up period to estimate incident dementia, but it is plausible that participants classified as incident cases may have experienced a long preclinical period with changes in brain anatomy and neuro-inflammatory biomarkers[113,114]. Because of the colocalization of CRP with amyloid-β plaques in brain parenchyma, and the correlation between CRP cerebrospinal fluid concentrations and peripheral levels, our results may be susceptible to reverse causation[45,115,116]. Future studies exploring longitudinal trajectories of inflammation with longer follow-up periods should address this limitation. Finally, in sensitivity models using the continuous log-transformation of CRP, we did not find statistical evidence of interaction or mediation. Further research should consider testing mediation and interaction at different cutoff points for CRP to understand if the relationship between racialization, systemic inflammation, and incident dementia is sensitive to different CRP thresholds[117].

Our analysis has several strengths, including quantifying the association between CRP and incident dementia in a large ($n = 6908$) and diverse sample of older adults in the United States. We had rich data on well-known confounding variables, including a major genetic risk factor for dementia. We also performed a sensitivity analysis for our decomposition models and obtained consistent estimates. Notably, treating *APOE-ε4* allele carrier status as a potential mediator-outcome confounder affected by the exposure did not alter our conclusions. However, this raises the question of the complex interrelation between the social construction of race through the racialization process, and genetic ancestry. The Health and Retirement Study correlated the genetic diversity of its sample to self-reported racialized social categories. However, this artifact of the data does not reflect genome-wide differences between racialized groups[118]. Additionally, research shows that individuals with African, Hispanic, and Caribbean ancestry have a higher frequency of the *APOE-ε4* allele than individuals with European ancestry. The higher frequency of the *ε4* allele does not confer individuals of African or Hispanic ancestry a higher risk for dementia as it does to individuals of European ancestry[76]. This poses the question of whether the observed variation in dementia risk among individuals from different ancestral populations is modified through the interplay between the *APOE-ε4* allele and biological factors such as systemic inflammation[74,119], which in turn is highly influenced by the racialization process. Finally, we also tested the degree to which an unmeasured confounder could nullify our indirect effects by calculating mediational E-values for our decomposition models. Nonetheless, our major strength is the novel application of a recently developed methodological approach that unifies mediation and racialized category interaction effects into one scientific query.

The results of this study may serve as empirical evidence for existing theoretical frameworks that seek to explain how racism is embodied in the physiology of the individuals who survive it, and how this embodiment affects their susceptibility to health and disease. The contextualization of race in causal methodology is part of an ongoing epidemiological debate. Our interpretation of disparities among racialized groups is up-to-date with

recent developments on structural racism and causal methodology[17,71]. Finally, this work has important implications for public health. We demonstrated that, in comparison to non-Hispanic white adults, minoritized racialized groups in the United States have elevated levels of systemic inflammation even after controlling for individual-level factors. Therefore, public health efforts should devote attention to understanding how structural racism and the process of racialization are associated with systemic inflammation in these populations, to ameliorate the racial gap in adverse cognitive outcomes.

## Data availability
Survey data are publicly available (https://hrs.isr.umich.edu/data-products), and genetic data were available through dbGaP (https://dbgap.ncbi.nlm.nih.gov; phs000428.v2.p2) and the National Institute on Aging Genetics of Alzheimer's Disease Data Storage Site (https://dss.niagads.org/; NG00119). Source data for Fig. 2 are available as Supplementary Data 1.

## Code availability
The code to produce these analyses is available in an accessible public repository (https://github.com/bakulskilab/Racialization_CRP_Dementia)[91].

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

## Acknowledgements

We would like to thank Dr. Paris B. Adkins-Jackson for her insightful comments and editorial support throughout this manuscript. We also thank the participants and staff of the Health and Retirement Study. This work was supported by the National Institutes of Health (grant numbers R01 AG055406, R01 AG067592, 3R01 AG067592-01S1, P30 AG072931, and R01 AG074887). The Health and Retirement Study is sponsored by the National Institute on Aging (U01 AG009740) and is conducted at the Institute for Social Research, University of Michigan. César Higgins Tejera was supported by 3R01 AG067592-01S1 and the University of Michigan Rackham Merit Fellowship program.

## Author contributions

C.H.T.: Conceptualization, study design, data curation, statistical analysis, writing, and editing. E.B.W.: Supervision, writing, and editing. M.T.H.: Writing

and editing. L.C.K.: Analytical support, writing, and editing. H.W.: Code review, writing, and editing. F.B.: Editing and writing of the initial draft. M.Z.: Statistical supervision and editing support. B.M.: Statistical supervision, writing, and editing. K.M.B.: Supervision of study design, statistical analysis, writing, editing, and funding.

## Competing interests

The authors declare no competing interests.
