## [Peer Review File · Communications Medicine]

Parts of this Peer Review File have been redacted as indicated to maintain the confidentiality of unpublished data.Reviewers' comments:

Reviewer #1 (Remarks to the Author):

The authors aimed to study the role of systemic inflammation as a mediator of racial/ethnic disparities in dementia and the role of race/ethnicity as a moderator of the association between systemic inflammation and dementia. The authors performed a secondary analysis of data from the Health and Retirement Study. The authors used causal mediation analysis to decompose the total racial/ethnic disparity in dementia into four components (controlled direct effect, pure natural indirect effect, mediated interaction, and reference interaction). Although I believe that this manuscript could be an important contribution to the literature, I do have some concerns regarding the analyses and interpretations of the results. Please find my point-by-point review below.

1. Dementia was determined using the Langa-Weir cut-off points to the composite score of the modified Telephone Interview for Cognitive Status (TICS). The way this is currently described in the manuscript makes it sound like all tests that were included in the TICS factored into the composite score (range 0-27). This is not correct, as the naming and orientation items are not included when determining dementia status using the Langa-Weir cut-off points. See also Crimmins et al. (2011).

Crimmins, E. M., Kim, J. K., Langa, K. M., & Weir, D. R. (2011). Assessment of cognition using surveys and neuropsychological assessment: the Health and Retirement Study and the Aging, Demographics, and Memory Study. *Journals of Gerontology Series B: Psychological Sciences and Social Sciences*, 66(suppl_1), i162-i171.

2. The authors excluded people with cognitive impairment (CIND) from the analyses. This means that selection into the analytical sample was dependent on the outcome variable. By excluding this group of participants who do not (yet) have dementia, the observed effects may be larger than the true effects. Based on the Langa-Weir cut-offs the participants with CIND do not have dementia. Therefore, I would suggest including the participants with cognitive impairment in the analyses by including them in the group of participants who do not have dementia. This could be followed by a sensitivity analysis in which the participants with CIND are included in the dementia group instead.

3. The authors choose to merge non-Hispanic Black and Hispanic participants into one group, which they termed the minoritized category. In addition, the authors performed subgroup analyses in which they compared 1) non-Hispanic Black and non-Hispanic White participants (i.e., excluding Hispanic participants), and 2) Hispanic White participants with non-Hispanic White participants (i.e., excluding non-Hispanic Black participants). It is unclear to me why the authors choose this strategy rather than treating race/ethnicity as a categorical variable (non-Hispanic White, non-Hispanic Black, and Hispanic). If race/ethnicity is treated as a categorical variable, comparisons could be made between all categories while preserving power. I believe that the CMAverse package also facilitates categorical (nominal) exposures, so software is not a limiting factor.

4. Race and ethnicity are not necessarily mutually exclusive categories. Were there any participants who identified as Hispanic Black? If so, how were these participants categorized?

5. The authors adjusted the analyses for multiple confounders. The authors mention that APOE may be an exposure-induced confounder of the mediator-outcome effect. Although this is explicitly stated for APOE, this may also be the case for some of the other confounders. For example, historically, people from minoritized groups had fewer educational opportunities compared to White people. It would be important to clarify whether any of the other confounders were affected by race/ethnicity, and if so, to treat these variables appropriately in the analyses.

6. The authors choose to dichotomize some of the categorical and continuous confounders (e.g., smoking, number of chronic health conditions). This may lead to residual confounding. Therefore, it would be more appropriate to include these variables as categorical or continuous rather than dichotomous.

7. Childhood socio-economic status could also be a potential confounder of the CRP-dementia association, as childhood socio-economic status has been found to be associated with both inflammation (e.g., see Milaniak & Jaffee, 2019) and dementia (e.g., see Cha, Farina, & Hayward, 2011). If possible, it would be important to adjust for (a proxy of) childhood socio-economic status in the analyses.

Cha, H., Farina, M. P., & Hayward, M. D. (2021). Socioeconomic status across the life course and dementia-status life expectancy among older Americans. *SSM-population health*, 15, 100921.

Milaniak, I., & Jaffee, S. R. (2019). Childhood socioeconomic status and inflammation: a systematic review and meta-analysis. *Brain, behavior, and immunity*, 78, 161-176.

8. Could the authors clarify their choice to dichotomize CRP?

9. The mediation analyses were performed using the CMAverse package for causal mediation analysis. This package includes six estimation methods. Could the authors clarify which method they used to estimate the effects? Could the authors also clarify what type of confidence intervals were estimated for the mediation effect estimates?

10. The mediational E-value was determined for the Total Natural Indirect Effect estimate, while this effect estimate is not displayed in Table 2. It would be helpful to report the Total Natural Indirect effect, so that it is clear to which the mediational E-value applies.

11. The authors conclude that the observed disparity in dementia was mediated by CRP. Although the mediated interaction was statistically significant, the estimate was small, and the estimate of the pure indirect effect was small and not significant. Table 2 shows that most of the disparity is explained by the direct effect (which includes the effects through other mediators than CRP) and by the reference interaction. Furthermore, the mediational E-value showed that an unmeasured confounder only needs to have a relatively small effect on CRP and dementia to explain away the mediated effect. Given that the authors did not adjust for childhood socioeconomic status and that there may be residual confounding, there is a possibility that the observed mediated effect is explained by confounding. Therefore, the language used in the conclusion of the abstract and in the discussion section is more conclusive than is supported by the results in the paper.

Reviewer #2 (Remarks to the Author):

The Mediating Role of Systemic Inflammation and Moderating Role of Race/Ethnicity in Racialized Disparities in Incident Dementia: A Decomposition Analysis

This study was a systematic analysis of whether C-reactive protein (CRP), a marker of systemic inflammation, mediates the relationship between race/ethnicity and dementia incidence or is moderated by race/ethnicity to predict racial disparities in incident dementia. Among minoritized participants (i.e., self-identified non-Hispanic Black and Hispanic Americans), high CRP was found to be associated with increased risk for dementia; the association was strongest for Hispanic Americans and was not significant for non-Hispanic Black Americans. CRP accounted for a small proportion (2%) of the relationship between minoritized (vs. non-minoritized) group and dementia incidence; however, the interaction between minoritized group status and increased CRP accounted for 12% of the disparity in dementia incidence. The moderating role of race/ethnicity was largely attributable to the Hispanic American group. The aims and findings of this study are very important because there has been little prior research investigating mechanisms linking racialization to health disparities in dementia risk. This study is one of the first to begin to undertake this complex issue. Questions and concerns about the manuscript are outlined below.

Introduction

1. To provide support for collapsing the 2 racialized categories (i.e., non-Hispanic Black and Hispanic) into one group, please describe how the deleterious consequences of racialization (i.e., the embodiment of racism) may operate similarly across marginalized groups.
2. Please explain in more detail why inflammation is a potential marker for racialization. What are

the pathways that link racialization to physiological and behavioral (e.g., coping) responses that increase inflammation?

3. A more thorough discussion of research on inflammation and dementia risk is needed. Hegazy et al, cited in the manuscript, found that low CRP was associated with greater risk for dementia. In another study (Gabin et al., 2018 doi: 10.1186/s12979-017-0106-3), CRP levels were positively associated with greater risk for dementia in younger old adults (60-70.5) but lower risk for dementia in older old adults (>70.6). Mixed findings from the literature should be addressed.

4. To provide additional support for testing CRP as a mediator, please briefly describe physiological mechanisms that may link increased inflammation and dementia risk.

5. The last sentence regarding APOE4 needs clarification; further elaboration should occur earlier in the introduction. It sounds as though an additional test will be performed to show that the mediation analysis, testing inflammation as a marker of racialization, is robust to influences from a major genetic risk factor for dementia (APOE4) which is also associated with lower CRP and more frequently found in the African American population. I think it would be important to note that within the African American population, APOE4 may not increase risk for AD as much as in the non-Hispanic White population (Tang et al., 1998).

6. The issue of APOE4 introduced in the last sentence raises the question of the potential overlap between racialized group status and shared genetic ancestry. Racial groups are social constructs and do not have any biological foundation; however, what if a certain proportion of the variance within some racialized groups is due to shared genetic ancestry? To what extent do processes of racialization interact with shared genetic ancestry, including and beyond APOE4? This is undoubtedly a very complex issue, but one that might require some recognition in the discussion section.

Methods

1. Please explain why 6-years was selected as the time frame for determining dementia incidence. The preclinical phase of Alzheimer's disease may last from 20-30 years. Participants who develop dementia within the 6-year window are already experiencing the physiological consequences of AD, which would likely include neuroinflammation. Can brain-derived CRP (Yasojima et al., 2000 doi.org/10.1016/S0006-8993(00)02970-X) increase plasma CRP levels? A longer time lag than 6-years would be needed to decrease influence from reverse causation. This limitation should be discussed.

2. Please explain what is meant by "any non-dementia baseline". How was cognitive status determined at baseline?

3. Regarding the correlation reported for plasma and blood spot CRP, was that result from a subsample of the HRS or from another study?

4. Because sex differences have been reported for CRP, please explain why sex was not tested as a moderator of racialized group status.

Results

1. Results that were not statistically significant should be reported as such and should be described in the discussion section as not statistically significant. This is in reference to statements that seem to imply an association, in spite of CIs including 0, e.g., "those with high CRP had only 1.09 (95% CI: 0.83, 1.43) times higher risk of dementia than those with low CRP" and "When decomposing the non-Hispanic Black vs non-Hispanic White disparity, we found that the mediating effect of CRP accounted for 2% (95% CI: -1%, 8%) of the disparity, and the portion attributable to the interaction accounted for 8% (95% CI: -5%, 21%)". And in the discussion: "When decomposing the non-Hispanic Black versus non-Hispanic White disparity, we observed that 8% was attributable to the interaction effect between non-Hispanic Black membership and high CRP."

Discussion

1. Even though the non-Hispanic Black group had the greatest odds for high CRP, CRP was not related to incident dementia in this group. This is an interesting finding that deserves greater elaboration.

2. The sample size of the non-Hispanic Black group was described as possibly too small to detect significant effects, but the size of the non-Hispanic Black group was slightly larger than the Hispanic group, in which significant effects were found. Was CRP highest in the non-Hispanic Black participants who dropped out?

3. Intersectionality (race x gender, race x education) may also play a role in dementia incidence

and should be discussed.

Reviewer #3 (Remarks to the Author):

The authors have written an interesting and important manuscript that evaluates the role of inflammation in understanding incident dementia across race/ethnic groups in the United States. They use the HRS data as well as the cognition file. Overall, I think it has great promise, and has the appropriate framing and methods, but requires some additional rationale or discussion to better place these findings in the larger literature.

The authors have indicated that they have excluded people who had CIND but never transitioned to having dementia. If you are looking at dementia incidence, I think that these people should remain in the denominator. I am wondering more about the justification of their removal, especially as minoritized populations are more likely to have CIND. Relatedly, as a sensitivity check, in order to not omit pertinent information, I might suggest the authors look at cognitive impairment events (whether it is the transition to CIND or Dementia). Lastly, some participants "recover" from dementia. I am wondering if the authors have explored back transitions (or if they also looked at how sensitive the definition of dementia event-- for example, dementia event could be based on having dementia status at two consecutive waves and/or death). Back transitions are common in the overall data. I am not sure about the specific subsample used in this study. I think this may be necessary if people who are misclassified may be experiencing infection or other health problems (that may be temporary) that will also elevate CRP. These suggestions do not need to go into the manuscript but would help in seeing how sensitive the models are to the definition of the dependent variable.

Additionally, it would be helpful to discuss the number of events per group since this will be driving the power of the model.

I am also wondering about the disciplinary norms of discussing findings that may not be significant. Several of the findings have coefficients with CIs that cross "0". I might suggest the authors amend the language to be cognizant that while the evidence is suggestive, it is not powerful enough to reject the null that it is not "0".

Lastly, I am wondering about the role of the racialization of Hispanics. I think the authors have done an admirable job discussing racialization, but I do think that the discussion or introduction would benefit from stating how this might be different for Black and Latinx older adults, and how different types of factors may be coming together to elevate CRP for both.

Overall, I think a few sensitivity checks and some minor changes to the manuscript would lead to a unique and important paper for the field.

We thank the editor for the opportunity to resubmit this manuscript. The reviewers provided incredibly thorough feedback, and while they highlighted the importance of this research, they identified important areas for improvement of the research. In particular, the reviewers suggested another operationalization of cognitive impaired participants. In response to reviewer feedback, we have successfully incorporated multiple sensitivity analyses into our manuscript and improved the rigor of multiple aspects of the paper. To be explicit about our operationalization of race, we also made a slight modification in our title to *“The Mediating Role of Systemic Inflammation and Moderating Role of Racialization in Disparities in Incident Dementia: A Decomposition Analysis”*. Our specific changes are described point by point below.

Reviewer # 1

General Comment:

The authors aimed to study the role of systemic inflammation as a mediator of racial/ethnic disparities in dementia and the role of race/ethnicity as a moderator of the association between systemic inflammation and dementia. The authors performed a secondary analysis of data from the Health and Retirement Study. The authors used causal mediation analysis to decompose the total racial/ethnic disparity in dementia into four components (controlled direct effect, pure natural indirect effect, mediated interaction, and reference interaction). Although I believe that this manuscript could be an important contribution to the literature, I do have some concerns regarding the analyses and interpretations of the results. Please find my point-by-point review below.

- **Response:** We thank the reviewer for their kind appreciation of this research manuscript.

Comment 1: Dementia was determined using the Langa-Weir cut-off points to the composite score of the modified Telephone Interview for Cognitive Status (TICS). The way this is currently described in the manuscript makes it sound like all tests that were included in the TICS factored into the composite score (range 0-27). This is not correct, as the naming and orientation items are not included when determining dementia status using the Langa-Weir cut-off points. See also Crimmins et al. (2011).

- **Response 1:** We thank the reviewers for this comment and agree that the language employed was confusing. To address this comment, we have modified the text under the methods section (outcome measure) in the following way:

“Cognitive status was evaluated at baseline and every two years through the Telephone Interview for Cognitive Status (TICS). Cognitive test results were recorded in a continuous scale including: 10 word immediate and delayed recall tests, a serial 7s subtraction test of working memory, counting backwards to assess attention and processing speed, an object naming to assess language, and recalling the date, president, and vice-president to assess orientation. All these items represent a cognitive functioning measure

that ranges from 0-35 points, with larger values indicating better cognitive performance. We used the Langa-Weir approach to classify HRS participants cognitive status based on a subset of the TICS assessments, specifically the immediate and delayed recall tests, serial 7s, and backwards counting. The range of scores of this subset of cognitive measures was 0-27 points.¹ According to the Langa-Weir approach, participants scoring 0-6 points were classified with dementia, 7-11 points were classified with CIND, and 12-27 points were classified as cognitively normal.¹ For the purpose of our analysis, we focused solely on participants who developed dementia over the 6-year study period, starting from cognitively normal or CIND at baseline. Proxy respondents were not included in our sample, as they did not provide blood spots.”

Comment 2: The authors excluded people with cognitive impairment (CIND) from the analyses. This means that selection into the analytical sample was dependent on the outcome variable. By excluding this group of participants who do not (yet) have dementia, the observed effects may be larger than the true effects. Based on the Langa-Weir cut-offs the participants with CIND do not have dementia. Therefore, I would suggest including the participants with cognitive impairment in the analyses by including them in the group of participants who do not have dementia. This could be followed by a sensitivity analysis in which the participants with CIND are included in the dementia group instead.

- **Response 2:** We thank the reviewer for this insightful comment and agree that excluding people with cognitive impairment non-dementia (CIND) from the analysis could result in selection bias. To address this issue, we have created a new analytic sample (N=6,908) in which participants with CIND at baseline or that developed CIND during the 6-year follow-up period were included as part of the reference group (i.e., cognitively normal or CIND). Because of this change in the analytical sample size, we updated our flowchart (**Supplemental Figure 1**), the descriptive (**Table 1 & Supplemental Tables 1 to 4**) and regression (**Table 2 & Supplemental Table 5 and 9**) tables; the mediation tables (**Table 3 & Supplemental Tables 6 to 8**), and all the figures throughout the manuscript (**Figure 1, 2 & Supplemental Tables 2-4**). Additionally, the cutoff point of C-reactive protein (CRP) was updated to - high $\geq 4.73\mu\text{g/mL}$ and low ($< 4.73\mu\text{g/mL}$), to represent the 75th percentile of the distribution of the new analytic sample of 6,908 participants. As suggested by the reviewer, we added relevant sensitivity analyses. We re-estimated incidence rate ratios using Poisson regression and mediation-interaction decomposition components but, this time, including CIND cases with the dementia group. We present these sensitivity analysis results in the results section and in the discussion.

In general, the incident rate ratios (IRR) estimated from the Poisson models with the new analytic sample size of 6,908 participants (CIND cases grouped with cognitively normal participants) were of similar magnitude to our initial analysis (dementia vs cognitively normal participants, and excluding CIND cases, N = 5,143). In the overall sample (N=6,098), and in fully adjusted models, high levels of CRP ($\geq 4.73\mu\text{g/mL}$) were associated with 1.23 (95%CI: 1.05, 1.44) times greater risk of 6-year incident dementia than low

CRP levels ($<4.73\mu\text{g/mL}$). This association was significant for Hispanic participants (IRR: 1.85; 95%CI: 1.27,2.70); marginally significant for non-Hispanic White participants (IRR: 1.19; 95%CI:0.98,1.45), null for non-Hispanic Black participants (IRR: 1.00; 95%CI: 0.72,1.37), and marginally significant for the minoritized group (non-Hispanic Black and Hispanic, IRR: 1.26, 95%CI: 0.98, 1.62). With respect to the mediation-interaction decomposition components, we found that when comparing the minoritized group to the most privileged group, elevated CRP mediated 3% (95%CI: 0%, 6%) of the racial disparity, and the proportion attributable to the interaction accounted for 14% (95%: 1%, 27%) of the racial disparity. For the non-Hispanic Black vs non-Hispanic White disparity, the mediation and interaction components were null. But, for the Hispanic vs non-Hispanic White disparity, we found that 28% (95%CI: 8%, 51%) of the disparity was accounted for by the interaction effect between the racialized social categories and the high levels of the biomarker. These results were of similar magnitude for the randomized analogue decomposition estimates.

We conducted appropriate changes throughout the methods, results, and discussion sections of the manuscript to accommodate this comment, as well as included the sensitivity analysis suggested by the reviewer.

Comment 3: The authors choose to merge non-Hispanic Black and Hispanic participants into one group, which they termed the minoritized category. In addition, the authors performed subgroup analyses in which they compared 1) non-Hispanic Black and non-Hispanic White participants (i.e., excluding Hispanic participants), and 2) Hispanic White participants with non-Hispanic White participants (i.e., excluding non-Hispanic Black participants). It is unclear to me why the authors choose this strategy rather than treating race/ethnicity as a categorical variable (non-Hispanic White, non-Hispanic Black, and Hispanic). If race/ethnicity is treated as a categorical variable, comparisons could be made between all categories while preserving power. I believe that the CMAverse package also facilitates categorical (nominal) exposures, so software is not a limiting factor.

- **Response 3:** We thank the reviewer for this comment and would like to provide clarity about our decision to group non-Hispanic Black and Hispanic participants into a unique group membership or minoritized status. This strategy, besides being a complementary approach to the suggestion to gain statistical power, reflects a current recommendation for the use of causal diagrams in the study of health disparity research.² This comment has directed us to be more explicit in our conceptualization of racialization as treatment assignment and to incorporate the historical processes that proceed racial designations in our causal diagram. Therefore, we have implemented the below changes in our methods section and our **Figure 1**.

Firstly, we would like to clarify that, in our study, the Hispanic group is composed of participants from different racial backgrounds. In the Health and Retirement Study questionnaire, participants are allowed to identify their race, followed by their ethnicity. This means that the Hispanic group category is

made-up of diverse racialized social groups. We have cross-tabulated the variables race and ethnicity in the complete sample of the HRS corresponding to waves 2006 and 2008 (the baseline years for our study). We can observe that nearly 60% of Hispanic participants identify themselves as White. However, nearly 40% of Hispanic participants identified themselves as belonging to “other” race, and only 3% of Hispanic participants identified themselves as Black. For instance, when we contrast Hispanic participants to their non-Hispanic White counterparts, we are comparing a heterogeneous group that in great proportion (around 40%) categorizes themselves as non-White. However, even if Hispanic participants self-identify as White, historically, and legally, Hispanic groups (i.e., Mexican Americans, Cubans, Puerto Ricans, Colombians) have been rarely treated as White citizens.^{3,4} In fact, in the United States, Hispanic groups have been subjected to unique experiences of discrimination due to their racialized status. For example, Hispanic individuals have been subjected to different forms of school and occupational segregation, and to a large extent, have been relegated to low-status positions due to their documentation status. Because throughout history, Hispanic individuals have encountered similar barriers than other oppressed racial groups (individuals racialized as Black and Indigenous peoples), in our study, we designated them as a minoritized category and therefore compared them to the most privileged racial group (participants racialized as non-Hispanic White).

REDACTED

Current recommendations for the study of racial disparities in epidemiologic research encourages researchers to contextualize race as a historical and socially constructed category by including a variable that reflects membership into a marginalized social designation (i.e., Hispanic, non-Hispanic Black, Indigenous people, etc.) and for comparison to a more privileged racialized social group (non-Hispanic White).² Based on this recommendation, we created our three racialized social categories, and because non-Hispanic White participants occupy the most privilege position in our society, they served as the reference category in our analysis. Further, non-Hispanic Black and Hispanic individuals may undergo different racialized treatment and experiences. These marginalized groups have undergone similar discriminatory practices.³⁻⁵ Since the aim of this study was to capture the effect of the racialization process on a biological pathway, contrasting two historically oppressed social groups would not address the main aim of our study.

Methods

Exposure

“Following current recommendations for the study of racial disparities in epidemiologic research,² we used participants’ self-reported racialized categories as a proxy measure of exposure to the racialization process. We compared each minoritized group (non-Hispanic Black and Hispanic) to the most privileged category (non-Hispanic White, reference group).² Participants racialized as Hispanic included those racialized as Hispanic White (58.6%), Hispanic Other race (37.1%), and Hispanic Black (2.8%). Although, non-Hispanic Black and Hispanic individuals are highly heterogeneous groups; in the United States, they have experienced structural discrimination in the form of redlining, educational segregation, mob violence, Jim Crow and anti-immigrant laws.^{3,5–10} These historical events have placed generations of non-Hispanic Black and Hispanic Americans behind their non-Hispanic White counterparts and are the root cause of important disparities in health and economic mobility.^{5,11–13} To capture this minoritized status and to leverage a larger sample size to detect small statistical effects, we combined non-Hispanic Black and Hispanic participants into a minoritized category. Throughout the manuscript, when comparing jointly non-Hispanic Black and Hispanic participants to the most privileged group, we use the terminology minoritized group; otherwise, we specify which racialized groups are being compared (i.e., non-Hispanic Black vs non-Hispanic White, or Hispanic vs non-Hispanic White). In our causal diagram (Figure 1), the arrow from the historical and institutional processes to the minoritized group membership indicates that these historical events force individual-level memberships into racially defined categories. These racialized social categories reflect hierarchies of privilege and social position rather than phenotypical, ancestral, or cultural attributes.¹⁴”

Figure 1: Directed Acyclic Graph illustrating the relationship between racialized social groups, systemic inflammation, and 6-year incident dementia in the Health and Retirement Study. Exposure (A) represents membership in a minoritized or racialized

group vs a privileged group (i.e., non-Hispanic Black and/or Hispanic participants vs non-Hispanic White participants). Racialized group membership stems from historical and structural processes related to racism and this forced membership status directly associated with incident dementia, as denoted in arrow c, and through systemic inflammation (Mediator M) as denoted by arrows (a) and (b). The association between systemic inflammation and incident dementia is denoted by arrow (b). However, the association between systemic inflammation and incident dementia can be modified by membership in minoritized racial status, as this model allows for exposure-mediator interaction. The set of confounders (C) account for exposure (A) - outcome (Y), exposure (A) - mediator (M), and mediator (M) - outcome (Y) confounders. This model also assumes that there are not mediator-outcome confounders (L) affected by the exposure (A).

Comment 4: Race and ethnicity are not necessarily mutually exclusive categories. Were there any participants who identified as Hispanic Black? If so, how were these participants categorized?

- **Response 4:** We thank the reviewer for this comment. In the previous response (response 3), we addressed how the Hispanic category was created. Briefly, in our study, the Hispanic category is made-up of individuals racialized as Hispanic White (58.6%), Hispanic Other race (37.1%), and to a lesser extent those racialized as Black (2.8%). We also explained why we studied the Hispanic group in relation to their non-Hispanic White counterpart, and why we created a unique category combining both, participants racialized as Hispanic and non-Hispanic Black.

Comment 5: The authors adjusted the analyses for multiple confounders. The authors mention that APOE may be an exposure-induced confounder of the mediator-outcome effect. Although this is explicitly stated for APOE, this may also be the case for some of the other confounders. For example, historically, people from minoritized groups had fewer educational opportunities compared to White people. It would be important to clarify whether any of the other confounders were affected by race/ethnicity, and if so, to treat these variables appropriately in the analyses.

- **Response 5:** We thank the reviewer for this comment and agree that there are other potential confounders that may be affected by the exposure in our analysis. Historically, educational attainment has been highly racialized and segregated in the United States. To follow the reviewer's comment, we treated the variable educational attainment in the analysis as a mediator-outcome confounder affected by the exposure. We performed this sensitivity analysis when comparing the 1) minoritized vs non-Hispanic White disparity, 2) the non-Hispanic Black vs non-Hispanic White disparity, and 3) the Hispanic vs non-Hispanic White disparity. These sensitivity mediation-interaction models included both, *APOE-ε4* allele carrier status, and educational attainment as variables that could be affected by the exposure (racialized social categories).

To this end, we used the CMAverse package to fit educational attainment as an ordered factor response and employed ordered logistic regression to model education attainment as a function of the exposure variable and other confounders. We have added this sensitivity analysis to the methods section and results. Overall, we found that when treating *APOE-ε4* and educational attainment as mediator-outcome confounders affected by the exposure, high levels of C-reactive protein mediated 3% (95%CI: 0%, 6%) of the minoritized vs non-Hispanic White disparity in incident dementia, and the interaction between minoritized status and high levels of C-reactive protein accounted for 15% (95%: 2%, 28%) of the observed racial disparities. This indicates that our main results were robust to this sensitivity check. We recognize that some health behaviors (body mass index, smoking, alcohol consumption) and multi-morbid conditions may be also mediator-outcome confounders affected by the exposure. However, because these variables were recorded at the same time of biomarker assessment, we think that these relationships should be explored in more detail in future work, in particular with mediation analyses that incorporate time ordered confounders and mediators. To clarify this to the reader we have added this limitation to the sections as stated below.

Methods

Sensitivity Analysis

*“[...] Furthermore, educational attainment has historically been racialized and segregated in the United States, and research has shown that this socio-economic health determinant is associated with both systemic inflammation and adverse cognitive outcomes.^{9,15–17} In an additional sensitivity randomized analogue model, we treated both educational attainment and *APOE-ε4* allele as mediator-outcome confounders affected by the exposure.. [...]”*

Results

Sensitivity Analysis

*“[...] Finally, because educational attainment has historically been racialized and segregated in the United States, and segregated schooling might be associated with both dementia, and C-reactive protein; we similarly conducted a randomized analogue mediation model to test the robustness of our mediated and interaction effects. This sensitivity model assumes that more than one confounder (i.e., *APOE-ε4* and educational attainment) was affected by the exposure (racialization process). We found that when comparing the minoritized group to the non-Hispanic White group, the mediating effect of CRP on incident dementia accounted for 3% (95% CI: 0%, 6%) of the disparity, and the proportion due to interaction accounted for 15% (95% CI: 2%, 28%) (**Supplemental Table 8**). We obtained similar estimates for the decomposition effects of the non-Hispanic Black vs non-Hispanic White disparity, and the Hispanic vs non-Hispanic White disparity than in the randomized analogue models that only included *APOE-ε4* as a sole mediator-*

outcome confounder affected by the exposure (Supplemental Table 8 & Supplemental Table 6) [...]”

Comment 6: The authors choose to dichotomize some of the categorical and continuous confounders (e.g., smoking, number of chronic health conditions). This may lead to residual confounding. Therefore, it would be more appropriate to include these variables as categorical or continuous rather than dichotomous.

- **Response 6:** We thank the reviewers for this comment and agree that dichotomizing categorical and continuous confounders could result in residual confounding bias. To follow the reviewer’s recommendations, we used the smoking variable in its three-level categorical form (current smoker, former smoker, and never smoker - the reference group). Similarly, the chronic health conditions variable was used in its continuous form (i.e., the number of chronic health conditions a participant’s reports, this number ranges from 0 to 8). We implemented these changes throughout our descriptive tables, regression models, and mediation-interaction decomposition models, and in text in the methods section (subsection covariates).

Comment 7: Childhood socio-economic status could also be a potential confounder of the CRP-dementia association, as childhood socio-economic status has been found to be associated with both inflammation (e.g., see Milaniak & Jaffee, 2019) and dementia (e.g., see Cha, farina, & Hayward, 2011). If possible, it would be important to adjust for (a proxy of) childhood socio-economic status in the analyses.

- **Response 7:** We thank the reviewers for this comment and bringing up to our attention the potential of confounding bias in our observational study. Although our models do not provide adjustment by childhood socio-economic status, research suggests that childhood socio-economic conditions are predictors of educational attainment. In previous responses we performed sensitivity analyses with models using educational attainment as a mediator-outcome confounder affected by the exposure. We found that our results were robust to this sensitivity check and given that educational attainment is strongly associated with dementia risk and C-reactive protein levels in our data, we believe that our results may be robust to this residual confounding bias. However, we acknowledge this limitation in our discussion, and use the references provided by the reviewer. Additionally, we reflect on the important relationship between health determinants such as childhood socio-economic position, and educational attainment and structural racism.

[...] Another important limitation is that our models did not adjust for childhood socio-economic status, and research suggests that this may be an important confounder between inflammation and dementia.^{18,19} However, in sensitivity models using APOE-ε4 and educational attainment as mediator-outcome confounders affected by the exposure, we did not observe significant changes in the magnitude of the mediating effect of systemic inflammation, or the moderating effect of the racialization process on the racial disparity. Our

*sensitivity analyses comparing minoritized status to the most privileged group yielded statistically significant results for the proportion due to mediation and interaction. These results were of similar magnitude to the main mediation models in which a potential violation of mediation analysis was ignored. However, it is noteworthy that important health determinants such as educational attainment, neighborhood characteristics, and childhood socio-economic status are, by a large degree, driven by historical and structural process that stem from racism.^{5,6,6} It is difficult to identify the indirect effect of systemic inflammation on the racial disparity in incident dementia without relying in the strong assumptions drawn in our causal diagram (**Figure 1**), and the temporal relationships between confounders and mediators. Structural racism through its multiple expressions is the root cause of economic disparities and physiological disruptions that may affect racialized individuals' susceptibility to disease.^{6,9,13} In this case, educational attainment, and other social health determinants (i.e., childhood socio-economic status) can be understood as mediator-outcome confounders affected by the exposure. Some of these health determinants are not included in our DAG and may be operating under the controlled direct effect of the racialization process. Racist policies and historical events such as redlining, mob violence, Jim Crow and anti-immigration laws have placed individuals racialized as non-Hispanic Black, Hispanic, and Indigenous at generational economic disadvantage and political underrepresentation. We argue that the cumulative effect of these disadvantages may have negative repercussion for the stress response with downstream consequences for cognitive aging. Our research framework is innovative, not in that we accounted for every possible confounding variable to identify the mediating effect of systemic inflammation on the racial disparity. But, in that we integrated a downstream biological determinant to understand the physiological underpinnings of the racialization process (i.e., the process of racializing individuals and differentially treating them across multiple domains of the social life).² Future research should expand on integrating structural measures of racism with biomarkers of disease to better capture the multiple biological expressions of racism, and its deleterious effects in human physiology.²⁰*

Comment 8: Could the authors clarify their choice to dichotomize CRP?

- **Response 8:** We thank the reviewer for this clarifying question. Clinical cut-off points for C-reactive protein have been developed for risk stratification of cardiovascular disease. The Center for Disease Control and Prevention (CDC) and the American Heart Association (AHA) established that blood concentration of C-reactive protein levels <1mg/L equated to a low risk for cardiovascular events; C-reactive protein concentrations between 1-3mg/L equated to an average risk, and C-reactive protein concentrations >3mg/L conferred a high risk for future cardiovascular events. To our knowledge, there are no current clinical cutoff points of C-reactive protein for risk stratification of dementia or cerebrovascular events. The highest quartile of C-reactive protein distribution (i.e., $\geq 75^{\text{th}}$) has been previously used for risk stratification of cerebrovascular outcomes, such as ischemic stroke and transient ischemic attack.²¹ Similarly, in our study, we used the highest quartile of the distribution

of C-reactive protein to denote high exposure to systemic inflammation. In our sample, the concentrations of C-reactive protein $\geq 75^{\text{th}}$ percentile equated to $\geq 4.73\mu\text{g/mL}$. This value is also within the cutoff threshold for high risk of cardiovascular events, as denoted by the CDC and the AHA. To clarify this to the reader we have added the following language to the methods section:

“[...] Because there are no clinical thresholds for risk stratification of CRP in blood and dementia incidence, we used the highest quartile of the distribution to denote exposure to high systemic inflammation levels. Previous studies have used the highest quartile of C-reactive protein to assess risk stratification of cerebrovascular events, such as ischemic stroke, and ischemic attack.²¹ In this study, we dichotomized CRP concentrations at the $\geq 75^{\text{th}}$ percentile (highest quartile, and blood concentrations $\geq 4.73\mu\text{g/mL}$) to explore its association with incident dementia, and its mediating effect of the racial disparity. These concentrations of CRP (i.e., $\geq 4.73\mu\text{g/mL}$) fall within the high stratification risk for cardiovascular events as suggested by the Center for Disease Control and Prevention (CDC) and the American Heart Association (AHA).²² [...]”

Comment 9: The mediation analyses were performed using the CMAverse package for causal mediation analysis. This package includes six estimation methods. Could the authors clarify which method they used to estimate the effects? Could the authors also clarify what type of confidence intervals were estimated for the mediation effect estimates?

- **Response 9:** We thank the reviewer for this clarifying question. The main mediation-interaction decomposition models estimates were obtained using the regression-based approach and direct counterfactual imputation estimation. We calculated 95% confidence intervals using bootstrap inference (i.e., 1000 boots in each mediation model), and set a random seed for replicability purposes. In the sensitivity mediation-interaction decomposition models, that is, the models that used *APOE-ε4* allele carrier status, and educational attainment as mediator-outcome confounders affected by the exposure, we used the g-formula approach and direct counterfactual imputation to estimate the decomposition effects. For these models, we also used bootstrap inference to calculate the 95% confidence intervals of each component of the decomposition. To clarify this to the reader, we have added the following text to our methods section.

Methods

Statistical Analysis

“[...] We performed four-way mediation-interaction decomposition analysis to evaluate whether CRP mediated disparities among racialized groups in incident dementia using the CMAverse R studio package, accounting for any interaction effect between minoritized group status and CRP.²³ This interaction

effect allowed us to capture whether belonging to a minoritized group differentially affected the strength of the association between systemic inflammation and incident dementia. Decomposition estimates were obtained using the `cmest` function of the `CMAverse` package, and we employed the regression-based approach and direct counterfactual imputation for estimation.²³ The 95% confidence intervals of our estimates were calculated using bootstrapping inference. We performed 1000 bootstraps in each procedure and set a random seed for reproducibility purposes. [...]

Methods

Sensitivity Analysis

[...] we employed randomized analogue models to test the robustness of our mediation analysis findings. In these models, decomposition estimates were obtained employing the `g`-formula approach and direct counterfactual imputation for estimation. The 95% confidence intervals of these randomized analogue model estimates were calculated using bootstrapping inference. We performed 1000 bootstraps in each procedure and set a random seed for reproducibility purposes. [...]

Comment 10: The mediational E-value was determined for the Total Natural Indirect Effect estimate, while this effect estimate is not displayed in Table 2. It would be helpful to report the Total Natural Indirect effect, so that it is clear to which the mediational E-value applies.

- **Response 10:** We thank the reviewer for this comment. We also think that the reviewer may be referring to table 3 instead of table 2. Table 3 shows the four elements of the mediation-interaction decomposition in the excess risk scale (i.e., controlled direct effect, the interaction reference, the interaction mediation, and the pure indirect effect) and the proportions attributable to each effect on the racial disparity. We reported the mediational E-value for the total natural direct effect, the total natural indirect effect, and their 95% confidence intervals in the **Supplemental Table 7A** (mediational E-values for decomposition models using the regression-based approach) and in the **Supplemental Table 7B** (mediational E-values for decomposition models using the `g`-formula approach). These natural effects are reported by the `CMAverse` package in the rate ratio scale. In these tables, and next to the natural direct and indirect effects, we reported the mediational E-value, and when applicable the E-value for the lower and upper bounds of the 95% confidence interval of the natural effects. We present these estimates for the minoritized vs non-Hispanic White disparity model, the non-Hispanic Black vs non-Hispanic White disparity model, and the Hispanic vs the non-Hispanic White disparity model.

Comment 11: The authors conclude that the observed disparity in dementia was mediated by CRP. Although the mediated interaction was statistically significant, the

estimate was small, and the estimate of the pure indirect effect was small and not significant. Table 2 shows that most of the disparity is explained by the direct effect (which includes the effects through other mediators than CRP) and by the reference interaction. Furthermore, the mediational E-value showed that an unmeasured confounder only needs to have a relatively small effect on CRP and dementia to explain away the mediated effect. Given that the authors did not adjust for childhood socioeconomic status and that there may be residual confounding, there is a possibility that the observed mediated effect is explained by confounding. Therefore, the language used in the conclusion of the abstract and in the discussion section is more conclusive than is supported by the results in the paper.

- **Response 11:** We thank the reviewer for this comment and believe that the reviewer has raised an important point for discussion. In our study we did not adjust for childhood socio-economic status, and as in any observational study, our analyses may be susceptible to residual confounding bias. However, some of the sensitivity checks that we have provided suggest that our results may be robust to this potential confounding bias. In the case of childhood socio-economic status, it may be important to contextualize this variable as a mediator-outcome confounder affected by the exposure (racialization), and the historical processes (i.e., slavery, mob violence, Jim Crow laws, redlining, anti-immigration laws) that relate to the racialization process. Just like in the case raised previously with educational attainment, childhood socio-economic status may be a variable affected by the racialization process and the historical events that took place during the childhood of the study participants. In our sensitivity analyses, we demonstrated that employing educational attainment as a mediator-outcome confounder did not nullify our mediational or interaction effects. Additionally, research has shown that childhood socio-economic status may be an important determinant of an individual's future educational attainment. Therefore, some of the effect of childhood socio-economic status may have imperfectly been captured by the educational attainment variable in our analysis. However, because these are important limitations of our study, we have explicitly stated so in the discussion section (as discussed in response 7) and encourage future research to accommodate time-ordered confounders to understand how early life exposure to economic hardship, and structural racism may influence adulthood inflammatory pathways and minoritized individuals' risk of dementia.

"[...] Another important limitation is that our models did not adjust for childhood socio-economic status, and research suggest that this may be an important confounder between inflammation and dementia.^{18,19} However, in a sensitivity models using APOE-ε4 and educational attainment as mediator-outcome confounders affected by the exposure, we did not observe significant changes in the magnitude of the mediating effect of systemic inflammation, or the moderating effect of the racialization process on the racial disparity. Our sensitivity models comparing minoritized status to the most privileged group yielded statistically significant results for the proportion due to mediation and interaction, these results were of similar magnitude to the main mediation models in which a potential violation of mediation analysis was ignored. However, it is noteworthy that important health determinants such as educational attainment, neighborhood characteristics, and childhood socio-

economic status are, by a large degree, driven by historical and structural process that stem from racism.^{5,6,6} It is difficult to identify the indirect effect of systemic inflammation on the racial disparity in incident dementia without relying in the strong assumptions drawn in our causal diagram (Figure 1), and the temporal relationships between confounders and mediator. Structural racism through its multiple expressions is the root cause of economic disparities and physiological disruptions that may affect racialized individuals' susceptibility to disease.^{6,9,13} In this case, educational attainment, and other social health determinants (i.e., childhood socio-economic status) can be understood as mediator-outcome confounders affected by the exposure. Some of these health determinants are not included in our DAG and may be operating under the controlled direct effect of the racialization process. Racist policies and historical events such as redlining, mob violence, Jim Crow and anti-immigration laws have placed individuals racialized as non-Hispanic Black, Hispanic, and Indigenous at generational economic disadvantage and political underrepresentation. We argue that the cumulative effect of these disadvantages may have negative repercussion for the stress response with downstream consequences for cognitive aging. Our research framework is innovative, not in that we accounted for every possible confounding variable to identify the mediating effect of systemic inflammation on the racial disparity. But, in that we integrated a downstream biological determinant to understand the physiological underpinnings of the racialization process (i.e., the process of racializing individuals and differentially treating them across multiple domains of the social life).^{2,24} Future research should expand on integrating structural measures of racism with biomarkers of disease to better capture the multiple biological expressions of racism, and its deleterious effects in human physiology.²⁰ Additionally, mediation analysis research should incorporate multiple biomarkers of systemic inflammation, and time-ordered confounders to better understand how early life-exposure to racism may influence systemic inflammation, and the cognitive trajectories of older adults. [...]"

Reviewer # 2

General Comment:

This study was a systematic analysis of whether C-reactive protein (CRP), a marker of systemic inflammation, mediates the relationship between race/ethnicity and dementia incidence or is moderated by race/ethnicity to predict racial disparities in incident dementia. Among minoritized participants (i.e., self-identified non-Hispanic Black and Hispanic Americans), high CRP was found to be associated with increased risk for dementia; the association was strongest for Hispanic Americans and was not significant for non-Hispanic Black Americans. CRP accounted for a small proportion (2%) of the relationship between minoritized (vs. non-minoritized) group and dementia incidence; however, the interaction between minoritized group status and increased CRP accounted for 12% of the disparity in dementia incidence. The moderating role of race/ethnicity was largely attributable to the Hispanic American group. The aims and findings of this study are very important because there has been little prior research investigating mechanisms linking racialization to

health disparities in dementia risk. This study is one of the first to begin to undertake this complex issue. Questions and concerns about the manuscript are outlined below.

- **Response:** We thank the reviewer for their assessment of our research and its potential impact, and for acknowledging the complexity that lies in linking structural racism to individual-level physiological processes.

Comment 1: To provide support for collapsing the 2 racialized categories (i.e., non-Hispanic Black and Hispanic) into one group, please describe how the deleterious consequences of racialization (i.e., the embodiment of racism) may operate similarly across marginalized groups.

- **Response 1:** We thank the reviewer for the opportunity to expand of pathways of embodiment among minoritized social groups. We have added the following sentence to the introduction section to explain how the deleterious consequences of racialization may operate similarly across marginalized groups and how individuals racialized as White uniquely benefit from racialization.

“[...] This study examines the mediating role of systemic inflammation, and the moderating role of racialized group on disparities in incident dementia in a large, diverse, population-based study.^{25,26} During European colonization, individuals were racialized based on skin tone, perceived country or continent of origin, and/or religious affiliation.²⁷ This categorization created social hierarchies where a privileged racialized group (i.e., non-Hispanic White) could receive the political and socioeconomic benefits at the expense marginalization of other groups.^{14,28–31} Because individuals racialized as White uniquely benefit from racialization, we examined their health benefits in comparison to other racialized groups^{3–5,11,12,28,32} We expected that a lack of compounded negatives experiences of discrimination, social exclusion, and marginalization was embodied as no chronic stress response or lower CRP.^{33–37} Research shows that persistent experiences of discrimination in minoritized individuals are associated with higher circulating levels of proinflammatory cytokines (i.e., C-reactive protein, Interleukin-6).^{38,39} This inflammatory state represents the pathway by which minoritized social groups embed the social exclusionary system in which they live. In this way systemic inflammation may be the central mechanism to understand minoritized individuals increased susceptibility to chronic conditions (i.e., cardiovascular disease, dementia, cancer) and early mortality.^{33–35,40,41} [...]”

Comment 2: Please explain in more detail why inflammation is a potential marker for racialization. What are the pathways that link racialization to physiological and behavioral (e.g., coping) responses that increase inflammation?

- **Response 2:** We thank the reviewer for this comment and the opportunity to expand on pathways that link racialization to physiological responses that increase inflammation. We have added the following paragraph to the introduction to address this comment.

“[...] In addition, systemic inflammation may explain racialized disparities in cognitive aging. The weathering hypothesis proposes that structural racism regularly activates the body’s stress-response causing minoritized racialized individuals to experience allostatic overload.^{42,43} The hyperactivation of the hypothalamus-pituitary-axis may lead to a chronic stress response characterized by elevated biomarkers of systemic inflammation (i.e., C-reactive protein, interleukin-6, tumoral necrosis factor-alpha), and stress hormones (i.e., cortisol, adrenaline).^{34,40,41,44} The inflammatory response is linked to the racialization process in that, through racialization theory one does not reduce racial discrimination to interpersonal forms of racism but recognizes that key features of structural racism are integrated in the process of assigning political value to fictional categorizations of race.^{10,14,29} For instance, disparities in biological markers of disease between racialized social groups are the mere physiological expressions of racism.⁴⁵ Therefore, systemic inflammation can be understood as the central mechanism linking the stress of racism to the racialized bodies of who survive it.⁴⁵ [...]”

Comment 3: A more thorough discussion of research on inflammation and dementia risk is needed. Hegazy et al, cited in the manuscript, found that low CRP was associated with greater risk for dementia. In another study (Gabin et al., 2018 doi: 10.1186/s12979-017-0106-3), CRP levels were positively associated with greater risk for dementia in younger old adults (60-70.5) but lower risk for dementia in older old adults (>70.6). Mixed findings from the literature should be addressed.

- **Response 3:** We thank the reviewer for this comment and the opportunity to address mixed findings of the relationship between C-reactive protein and dementia incidence in the literature. We have added the following sentence to the introduction to clarify this important point.

“[...] In a large cohort of individuals racialized as White of Danish descent, after adjusting for plasma lipids, health behaviors and the genetic influence of APOE-ε4 allele carrier status, low peripheral levels of CRP were associated with higher risk of Alzheimer’s disease and all cause dementia.⁴⁶ Other studies in European populations have found that high circulating levels of CRP were associated with higher risk of dementia.⁴⁷⁻⁴⁹ A recent study in Norway demonstrated that elevated levels of CRP may be associated with higher risk of dementia in adults of 60 to 70.5 years of age, but this association shifted for senior adults (>70.6 years).⁵⁰ These conflicting findings suggest that the relationship between circulating levels of CRP and dementia risk is complex and modified by factors like age⁵⁰ and cognitive domain,⁵¹ and therefore, large studies in diverse populations are warranted.^{11,42,43} Some research has shown that non-Hispanic Black women have the highest levels of CRP in comparison to non-Hispanic White women and men, and even non-Hispanic Black men.^{39,52,53} Thus, there is reason to believe that systemic

inflammation, via elevated CRP, may be important in linking the downstream effects of racialization to systemic inflammation and cognitive function.^{33,42} [...]”

Comment 4: To provide additional support for testing CRP as a mediator, please briefly describe physiological mechanisms that may link increased inflammation and dementia risk.

- **Response 4:**

“[...] The co-localization of glia and pro-inflammatory cytokines in amyloid- β plaques implicate neuro-inflammation has an important role in the pathogenesis of dementia.^{54–56} The neurodegenerative process that follows the extracellular deposits of amyloid- β peptide, the activation of the glial, and production of pro-inflammatory cytokines suggest that inflammation may be the result of a reaction to the abnormal accumulation of proteins in brain parenchyma.^{54,57} However, mounting evidence from observational epidemiological studies suggests a link between systemic inflammation and dementia onset.^{48,49,54,58} Pro-inflammatory cytokines can induce permeabilization of the brain blood barrier endothelium, inducing paracrine signaling with surrounding macrophages, and activation of the microglia.⁵⁶ Therefore, increasing epidemiological evidence suggests that systemic inflammation may be a driving force in the chain of events that lead to the onset of dementia.^{54,55,59–61} [...]”

Comment 5: The last sentence regarding APOE4 needs clarification; further elaboration should occur earlier in the introduction. It sounds as though an additional test will be performed to show that the mediation analysis, testing inflammation as a marker of racialization, is robust to influences from a major genetic risk factor for dementia (APOE4) which is also associated with lower CRP and more frequently found in the African American population. I think it would be important to note that within the African American population, APOE- ϵ 4 may not increase risk for AD as much as in the non-Hispanic White population (Tang et al., 1998).

- **Response 5:** We thank the reviewer for this comment and have modified our introduction to accommodate the differential cumulative risk conferred by the APOE- ϵ 4 allele across population groups with different ancestry. We also reflect on our choice to use randomized analogue models as sensitivity analysis checks to test the robustness of our mediation models.

“[...] Finally, because apolipoprotein E (APOE) is associated with lower circulating levels of CRP (our mediator),^{62,63} and the carrier status of the APOE- ϵ 4 allele confers a different cumulative risk for the development of dementia in individuals of African, Hispanic, and European ancestry.⁶⁴ We used randomized analogue models, in sensitivity analyses, to test whether

APOE-ε4 allele could be better treated as a mediator-outcome confounder affected by the exposure (racialized social categories) rather than solely as a confounder. Although genetic ancestry can have important effects in human health, its effects are distinct from the social construction of race.⁵⁰ However, in the Health and Retirement Study, racialized social categories are artificially paired to genetic ancestry and this feature of the data represented an opportunity to test the robustness of our main mediation analysis. In addition, we treated educational attainment as another potential mediator-outcome confounder affected by the exposure, given that in the United States educational attainment has been highly segregated, and education is an important factor associated with dementia onset.⁹ We revised these sensitivity analyses in light of our main mediation models, and provided a comprehensive conceptualization for the use of C-reactive protein as a potential pathway to understand disparities in incident dementia among racialized social groups. [...]

Comment 6: The issue of APOE4 introduced in the last sentence raises the question of the potential overlap between racialized group status and shared genetic ancestry. Racial groups are social constructs and do not have any biological foundation; however, what if a certain proportion of the variance within some racialized groups is due to shared genetic ancestry? To what extent do processes of racialization interact with shared genetic ancestry, including and beyond APOE4? This is undoubtedly a very complex issue, but one that might require some recognition in the discussion section.

- **Response 6:** We thank the reviewer for this comment and have expanded on the issue of racialization, genetic ancestry, APOE-e4 carrier status, and dementia risk in the discussion section.

[...] Notably, treating APOE-ε4 allele carrier status as potential mediator-outcome confounder affected by the exposure did not alter our conclusions. However, this raises the question of the complex interrelation between the social construction of race through the racialization process, and genetic ancestry. The Health and Retirement Study correlated the genetic diversity of its sample to self-reported racialized social categories. However, this artifact of the data does not reflect genome-wide differences between racialized groups.⁶⁵ Additionally, research shows that individuals with African, Hispanic, and Caribbean ancestry have a higher frequency of the APOE-ε4 allele than individuals with European ancestry. The higher frequency of the ε4 allele does not confer individuals of African or Hispanic ancestry a higher risk for dementia as it does to individuals of European ancestry.⁶⁴ This poses the question of whether the observed variation in dementia risk among individuals from different ancestral populations is modified through the interplay between the APOE-ε4 allele and biological factors such as systemic inflammation,^{62,66} which in turn is highly influenced by the racialization process. [...]

Comment 7: Please explain why 6-years was selected as the time frame for determining dementia incidence. The preclinical phase of Alzheimer's disease may last from 20-30 years. Participants who develop dementia within the 6-year window are already experiencing the physiological consequences of AD, which would likely include neuroinflammation. Can brain-derived CRP (Yasojima et al., 2000 [doi.org/10.1016/S0006-8993\(00\)02970-X](https://doi.org/10.1016/S0006-8993(00)02970-X)) increase plasma CRP levels? A longer time lag than 6-years would be needed to decrease influence from reverse causation. This limitation should be discussed.

- **Response 7:** We thank the reviewer for the opportunity to explain our time frame for determining dementia incidence. We acknowledged that Alzheimer's prodromal and preclinical phase window has an average of 20 years, while the average clinical duration of the disease is 8-10 years and agree that due to our short follow-up period, we are unable to rule out reverse causation. To address this limitation, we have added the following sentence to the discussion section.

"[...] Lastly, during the prodromal phase of Alzheimer's disease (20 years) there are changes in cerebrospinal fluid concentrations of Amyloid- β 42 and other inflammatory biomarkers that are predictive of disease onset.⁶⁷ In large observational studies like the Health and Retirement Study, participants are routinely screened for changes in their cognitive function; clinical impairment debuts with changes in cognitive tests scores starting approximately 6 years before symptoms onset.⁶⁷ We used a 6-year follow-up period to estimate incident dementia, but it is plausible that participants classified as incident cases may have experienced a long prodromal period with changes in brain anatomy and neuro-inflammatory biomarkers. Because of the colocalization of CRP with amyloid- β plaques in brain parenchyma, and the correlation between CRP cerebrospinal fluid concentrations and peripheral levels, our results may be susceptible to reverse causation.^{46,68,69} Future studies exploring longitudinal trajectories of inflammation with longer follow-up periods should address this limitation. [...]"

Comment 8: Please explain what is meant by "any non-dementia baseline". How was cognitive status determined at baseline?

- **Response 8:** We thank the reviewer for this clarifying question and have addressed in the methods section with the following sentence:

"For the purpose of our analysis we focused solely on participants who did not have dementia at baseline (i.e., cognitively normal, or CIND) and who developed dementia over the 6-year study period."

Comment 9: Regarding the correlation reported for plasma and blood spot CRP, was that result from a subsample of the HRS or from another study?

- **Response 9:** Thank you for the opportunity to clarify the CRP measurement approaches. Validation of the laboratory approach for blood spot CRP

measures was demonstrated in a previous independent sample with collection procedures designed to mimic the Health and Retirement Study. In the independent validation study, CRP concentrations from dried blood spot samples were linearly related to concentrations in plasma samples ($n = 87$ paired samples, Pearson $R = 0.99$).⁷⁰ We revised the methods as follows:

Circulating CRP was measured in blood spots using an enzyme-linked immunosorbent assay (ELISA).⁷⁰ The CRP assay lower limit of detection was 0.035mg/L, the within-assay imprecision was 8.1%, and the between-assay imprecision was 11.0%. In an independent sample, this dried blood spot approach was validated against the more typical plasma sample measures ($n = 87$ paired samples, Pearson $R = 0.99$).⁷⁰

Comment 10: Because sex differences have been reported for CRP, please explain why sex was not tested as a moderator of racialized group status.

- **Response 10:** We agree that this would be an important research question and we thank the reviewer for the opportunity to clarify why an interaction effect between race and sex was not tested. As shown in our distributional tables (**Supplemental Table 2 & Supplemental Table 3**), we likely lack the statistical power to detect meaningful difference in a race and sex stratified analysis. For example, the sample size for minoritized men of color were small. Our study had only $n=260$ men racialized as non-Hispanic Black, and $n=230$ men racialized as Hispanic. An analysis exploring the interaction effect between sex and race would require a larger sample size to detect statistically meaningful effects, in particular, because the mediated effect of CRP protein in the racial disparity was of small magnitude.

Comment 11: Results that were not statistically significant should be reported as such and should be described in the discussion section as not statistically significant. This is in reference to statements that seem to imply an association, in spite of CIs including 0, e.g., “those with high CRP had only 1.09 (95% CI: 0.83, 1.43) times higher risk of dementia than those with low CRP” and “When decomposing the non-Hispanic Black vs non-Hispanic White disparity, we found that the mediating effect of CRP accounted for 2% (95% CI: -1%, 8%) of the disparity, and the portion attributable to the interaction accounted for 8% (95% CI: -5%, 21%)”. And in the discussion: “When decomposing the non-Hispanic Black versus non-Hispanic White disparity, we observed that 8% was attributable to the interaction effect between non-Hispanic Black membership and high CRP.”

- **Response 11:** We thank the reviewers for their comment on the statistical significance of our results. we have reported not statistically significant results as such in both the results and the discussion section.

Comment 12: Even though the non-Hispanic Black group had the greatest odds for high CRP, CRP was not related to incident dementia in this group. This is an interesting finding that deserves greater elaboration.

- **Response 12:** We thank the reviewer for this comment and have provided the following sentences to better explain these findings.

[...] We found that high levels of systemic inflammation were associated with incident dementia in the overall sample, in the minoritized group, the Hispanic group, and the non-Hispanic White group. Although participants racialized as non-Hispanic Black had the highest levels of systemic inflammation, among this group, elevated CRP protein was not associated with incident dementia. An explanation for this finding may be that adults racialized as non-Hispanic Black exhibited high systemic inflammation levels as a result of the high-effort of coping against the stress of racism, but this translated only superficially into changes in their cognitive tests scores, a syndrome known in the literature as John Henryism.⁷¹⁻⁷³ [...]

Comment 13: The sample size of the non-Hispanic Black group was described as possibly too small to detect significant effects, but the size of the non-Hispanic Black group was slightly larger than the Hispanic group, in which significant effects were found. Was CRP highest in the non-Hispanic Black participants who dropped out?

- **Response 13:** We thank the reviewer for this comment. An internal analysis of our data revealed that among those who had missing information, the distribution of mean CRP levels is as followed:
 - Non-Hispanic Black (n=244) & CRP = 7.41
 - Hispanic (n=161) & CRP = 5.20
 - Non-Hispanic White (n=1279) & CRP = 5.06

Therefore, we can confirm that among participants who were dropped out from our study, those racialized as non-Hispanic Black had the highest levels of CRP. As hypothesized in our discussion, these results suggest that:

“[...] Chronic systemic inflammation may predispose Black participants for other competing events such as diabetes, cardiovascular disease, stroke, and premature death,⁷⁴⁻⁷⁶ which in turn may affect Black participants’ likelihood of retention during the study period. Although our models controlled for confounding bias by these potential competing events, we did not account for selection bias issues in our analysis, and future research should inform how differential loss to follow-up affects the relationship between systemic inflammation and dementia in Black participants. [...]”

Comment 14: Intersectionality (race x gender, race x education) may also play a role in dementia incidence and should be discussed.

- **Response 14:** We agree and thank the reviewer for this comment and the opportunity to discuss the how intersectionality should be integrated into

future research. To address this comment, we added the following sentence to the discussion section.

“[...] And, because participants at the intersection of multiple marginalized identities (i.e., non-Hispanic Black women and Hispanic women) exhibited higher levels of systemic inflammation, future work should characterize the role of racism and sexism in inflammation trajectories and dementia risk. [...]”

Reviewer # 3

General Comment: The authors have written an interesting and important manuscript that evaluates the role of inflammation in understanding incident dementia across race/ethnic groups in the United States. They use the HRS data as well as the cognition file. Overall, I think it has great promise, and has the appropriate framing and methods, but requires some additional rationale or discussion to better place these findings in the larger literature.

- **Response:** We thank the reviewer for the assessment of our manuscript and welcome their invitation to expand on our rationale with respect to our methods and the operationalization of the racialized social categories

Comment 1: The authors have indicated that they have excluded people who had CIND but never transitioned to having dementia. If you are looking at dementia incidence, I think that these people should remain in the denominator. I am wondering more about the justification of their removal, especially as minoritized populations are more likely to have CIND. Relatedly, as a sensitivity check, in order to not omit pertinent information, I might suggest the authors look at cognitive impairment events (whether it is the transition to CIND or Dementia). Lastly, some participants "recover" from dementia. I am wondering if the authors have explored back transitions (or if they also looked at how sensitive the definition of dementia event-- for example, dementia event could be based on having dementia status at two consecutive waves and/or death). Back transitions are common in the overall data. I am not sure about the specific subsample used in this study. I think this may be necessary if people who are misclassified may be experiencing infection or other health problems (that may be temporary) that will also elevate CRP. These suggestions do not need to go into the manuscript but would help in seeing how sensitive the models are to the definition of the dependent variable.

- **Response 1:** We thank the reviewer for this comment and have newly addressed some of these issues throughout the manuscript. There are other important comments, such as back transitions from dementia to CIND "recovering" that can be explored in future research. First, we included participants who experienced cognitive impairment non-dementia (CIND) in the denominator. We re-estimated all our descriptive, regression, and mediation analyses, and presented the updated results throughout the manuscript. As a sensitivity analysis, we included participants with CIND in

the dementia groups and re-estimated all of the models. Having participants with CIND in the denominators did not change the results in our main models, or in the mediation analysis. However, when participants with CIND were included in the numerator, our estimates became null. This indicates that CRP may have a more important effect in differentiating dementia cases from cognitive impairment non-dementia. The suggestion of adding CIND participants into the denominator was also given by reviewer # 1, and we have copied our respond below as it directly addresses this comment. We believe that back transition events, as in the case of participants who were classified as having dementia in a particular wave, but were reclassified as cognitively normal or CIND in a future waves, deserves its own attention, and we do not want to extend misclassification of incident dementia cases that may have biased our current results. Future research in the Health and Retirement Study should clarify what an incident dementia case is and how to treat participants who experience transition from a cognitively impaired state to a cognitively normal one.

“Response 2 to reviewer # 1: We thank the reviewer for this insightful comment and agree that excluding people with cognitive impairment non-dementia (CIND) from the analysis could result in selection bias. To address this issue, we have created a new analytic sample (N=6,908) in which participants with CIND at baseline or that developed CIND during the 6-year follow-up period were included as part of the reference group (i.e., cognitively normal or CIND). Because of this change in the analytical sample size, we updated our flowchart, the descriptive and regression tables; the mediation tables, and all the figures throughout the manuscript. Additionally, the cutoff point of C-reactive protein (CRP) was updated to high $\geq 4.73\mu\text{g/mL}$ and low ($< 4.73\mu\text{g/mL}$), to represent the 75th percentile of the distribution of the new analytic sample of 6,908 participants. As suggested by the reviewer, we added a sensitivity analyses. We re-estimated incidence rate ratios using Poisson regression and mediation-interaction decomposition components but, this time, including CIND cases with the dementia group. We present these sensitivity analysis results in the results section and in the discussion.

In general, the incident rate ratios (IRR) estimated from the Poisson models with the new analytic sample size of 6,908 participants (CIND cases grouped with cognitively normal participants) were of similar magnitude to our initial analysis (dementia vs cognitively normal participants, and excluding CIND cases, N = 5,143). In the overall sample (N=6,098), and in fully adjusted models, high levels of CRP ($\geq 4.73\mu\text{g/mL}$) were associated with 1.23 (95%CI: 1.05, 1.44) time greater risk of 6-year incident dementia than low CRP levels ($< 4.73\mu\text{g/mL}$). This association was significant for Hispanic participants (IRR: 1.85; 95%CI: 1.27, 2.70); marginally significant for non-Hispanic White participants (IRR: 1.19; 95%CI: 0.98, 1.45), null for non-Hispanic Black participants (IRR: 1.00; 95%CI: 0.72, 1.37), and marginally significant for the minoritized group (non-Hispanic Black and Hispanic, IRR: 1.26, 95%CI: 0.98, 1.62). With respect to the mediation-interaction decomposition components, we found that when comparing the minoritized group to the most privileged group, elevated CRP mediated 3% (95%CI: 0%, 6%) of the racial disparity, and the proportion attributable to the interaction accounted for 14% (95%: 1%,

27%) of the racial disparity. For the non-Hispanic Black vs non-Hispanic White disparity, the mediation and interaction components were null. But, for the Hispanic vs non-Hispanic White disparity, we found that 28% (95%CI: 8%,51%) of the disparity was accounted for by the interaction effect between the racialized social categories and the high levels of the biomarker. These results were of similar magnitude for the randomized analogues decomposition estimates.

We conducted appropriate changes through the methods, results, and discussion sections of the manuscript to accommodate this comment, and the sensitivity analysis suggested by the reviewer.”

Comment 2: Additionally, it would be helpful to discuss the number of events per group since this will be driving the power of the model.

- **Response 2:** We thank you for this opportunity. We agree that low number of events per racialized social group can hinder our ability to detect statistically significant associations and have added the following sentence to the discussion.

“[...] Although participants racialized as non-Hispanic Black had the highest levels of systemic inflammation, elevated CRP protein was not associated with incident dementia. An explanation for this finding may be that adults racialized as non-Hispanic Black exhibit high systemic inflammation levels as a result of high-effort coping against the stress of racism but this translated only superficially into changes in their cognitive tests scores, a syndrome known in the literature as John Henryism.⁷¹⁻⁷³ Additionally, the majority of incident dementia cases occurred in participants racialized as non-Hispanic White (n=521 or 65.5%), and to a lesser extent in participants racialized as non-Hispanic Black (n=171 or 21%) and Hispanic (n=103, 13%), the lower number of events in minoritized participants also suggest that statistical power issues may be a limitation [...].”

Comment 3: I am also wondering about the disciplinary norms of discussing findings that may not be significant. Several of the findings have coefficients with CIs that cross "0". I might suggest the authors amend the language to be cognizant that while the evidence is suggestive, it is not powerful enough to reject the null that it is not "0".

- **Response 3:** We thank the reviewer for this comment, and this issue has also been raised by reviewer # 2 in comment 11. We have referred to results that are not statistically significant as such throughout the results and discussion section of the manuscript.

Comment 4: Lastly, I am wondering about the role of the racialization of Hispanics. I think the authors have done an admirable job discussing racialization, but I do think

that the discussion or introduction would benefit from stating how this might be different for Black and Latinx older adults, and how different types of factors may be coming together to elevate CRP for both.

- **Response 4:** We thank the reviewer for this comment and the opportunity to expand on the process of racialization and how it may similarly affect marginalized racial groups in the US. Reviewer # 2 also made a similar comment, and we addressed in the following way:

“[...] This study examines the mediating role of systemic inflammation, and the moderating role of racialized group on disparities in incident dementia in a large, diverse, population-based study.^{25,26} During European colonization, individuals were racialized based on skin tone, perceived country or continent of origin, and/or religious affiliation.²⁷ This categorization created social hierarchies where a privileged racialized group (i.e., non-Hispanic White) could receive the political and socioeconomic benefits at the expense marginalization of other groups.^{14,28–31} Because individuals racialized as White uniquely benefit from racialization, we examined their health benefits in comparison to other racialized groups^{3–5,11,12,28,32} We expected that a lack of compounded negatives experiences of discrimination, social exclusion, and marginalization was embodied as no chronic stress response or lower CRP.^{33–37} Research shows that persistent experiences of discrimination in minoritized individuals are associated with higher circulating levels of proinflammatory cytokines (i.e., C-reactive protein, Interleukin-6).^{38,39} This inflammatory state represents the pathway by which minoritized social groups embed the social exclusionary system in which they live. In this way systemic inflammation may be the central mechanism to understand minoritized individuals increased susceptibility to chronic conditions (i.e., cardiovascular disease, dementia, cancer) and early mortality.^{33–35,40,41} [...]”

References

1. Crimmins EM, Kim JK, Langa KM, Weir DR. Assessment of Cognition Using Surveys and Neuropsychological Assessment: The Health and Retirement Study and the Aging, Demographics, and Memory Study. *J Gerontol B Psychol Sci Soc Sci*. 2011;66B(Suppl 1):i162-i171. doi:10.1093/geronb/gbr048
2. Howe CJ, Bailey ZD, Raifman JR, Jackson JW. Recommendations for Using Causal Diagrams to Study Racial Health Disparities. *American Journal of Epidemiology*. 2022;191(12):1981-1989. doi:10.1093/aje/kwac140
3. Ortiz V, Telles E. Racial Identity and Racial Treatment of Mexican Americans. *Race Soc Probl*. 2012;4(1):41-56. doi:10.1007/s12552-012-9064-8
4. Carrigan WD, Webb C. *Forgotten Dead: Mob Violence Against Mexicans in the United States, 1848-1928*. OUP USA; 2013.
5. Bailey ZD, Feldman JM, Bassett MT. How Structural Racism Works — Racist Policies as a Root Cause of U.S. Racial Health Inequities. Malina D, ed. *N Engl J Med*. 2021;384(8):768-773. doi:10.1056/NEJMms2025396
6. Bailey ZD, Krieger N, Agénor M, Graves J, Linos N, Bassett MT. Structural racism and health inequities in the USA: evidence and interventions. *The Lancet*. 2017;389(10077):1453-1463. doi:10.1016/S0140-6736(17)30569-X
7. Almeida J, Biello KB, Pedraza F, Wintner S, Viruell-Fuentes E. The association between anti-immigrant policies and perceived discrimination among Latinos in the US: A multilevel analysis. *SSM - Population Health*. 2016;2:897-903. doi:10.1016/j.ssmph.2016.11.003

8. Ryabov I, Van Hook J. School segregation and academic achievement among Hispanic children. *Social Science Research*. 2007;36(2):767-788. doi:10.1016/j.ssresearch.2006.04.002
9. Adkins-Jackson PB, Weuve J. Racially Segregated Schooling and the Cognitive Health of Black Adults in the United States—Why It Matters. *JAMA Network Open*. 2021;4(10):e2130448. doi:10.1001/jamanetworkopen.2021.30448
10. Omi M, Winant H. *Racial Formation in the United States*. Routledge; 2014.
11. Krieger N. Methods for the Scientific Study of Discrimination and Health: An Ecosocial Approach. *Am J Public Health*. 2012;102(5):936-944. doi:10.2105/AJPH.2011.300544
12. Smedley A, Smedley BD. Race as biology is fiction, racism as a social problem is real: Anthropological and historical perspectives on the social construction of race. *American Psychologist*. 2005;60(1):16-26. doi:10.1037/0003-066X.60.1.16
13. Adkins-Jackson PB, George KM, Besser LM, et al. The structural and social determinants of Alzheimer's disease related dementias. *Alzheimer's & Dementia*. n/a(n/a). doi:10.1002/alz.13027
14. Bonilla-Silva E. It's not the rotten apples! Why family scholars should adopt a structural perspective on racism. *Journal of Family Theory & Review*. n/a(n/a). doi:10.1111/jftr.12503
15. Petersen KL, Marsland AL, Flory J, Votruba-Drzal E, Muldoon MF, Manuck SB. Community Socioeconomic Status is Associated With Circulating Interleukin-6 and C-Reactive Protein. *Psychosomatic Medicine*. 2008;70(6):646. doi:10.1097/PSY.0b013e31817b8ee4
16. Owen N, Poulton T, Hay FC, Mohamed-Ali V, Steptoe A. Socioeconomic status, C-reactive protein, immune factors, and responses to acute mental stress. *Brain, Behavior, and Immunity*. 2003;17(4):286-295. doi:10.1016/S0889-1591(03)00058-8
17. Reuser M, Willekens FJ, Bonneux L. Higher education delays and shortens cognitive impairment. A multistate life table analysis of the US Health and Retirement Study. *Eur J Epidemiol*. 2011;26(5):395-403. doi:10.1007/s10654-011-9553-x
18. Milaniak I, Jaffee SR. Childhood socioeconomic status and inflammation: A systematic review and meta-analysis. *Brain Behav Immun*. 2019;78:161-176. doi:10.1016/j.bbi.2019.01.018
19. Cha H, Farina MP, Hayward MD. Socioeconomic status across the life course and dementia-status life expectancy among older Americans. *SSM Popul Health*. 2021;15:100921. doi:10.1016/j.ssmph.2021.100921
20. Adkins-Jackson PB, Incollingo Rodriguez AC. Methodological approaches for studying structural racism and its biopsychosocial impact on health. *Nursing Outlook*. 2022;70(5):725-732. doi:10.1016/j.outlook.2022.07.008
21. Plasma Concentration of C-Reactive Protein and Risk of Ischemic Stroke and Transient Ischemic Attack. doi:10.1161/hs1101.098151

22. Salazar J, Martínez MS, Chávez M, et al. C-Reactive Protein: Clinical and Epidemiological Perspectives. *Cardiology Research and Practice*. 2014;2014:1-10. doi:10.1155/2014/605810
23. Shi B, Choirat C, Coull BA, VanderWeele TJ, Valeri L. CMAverse: A Suite of Functions for Reproducible Causal Mediation Analyses. *Epidemiology*. 2021;32(5):e20. doi:10.1097/EDE.0000000000001378
24. Graetz N, Boen CE, Esposito MH. Structural Racism and Quantitative Causal Inference: A Life Course Mediation Framework for Decomposing Racial Health Disparities. *J Health Soc Behav*. Published online January 8, 2022:002214652110661. doi:10.1177/00221465211066108
25. Barnes LL. Biomarkers for Alzheimer Dementia in Diverse Racial and Ethnic Minorities-A Public Health Priority. *JAMA Neurol*. 2019;76(3):251-253. doi:10.1001/jamaneurol.2018.3444
26. Wilkins CH, Schindler SE, Morris JC. Addressing Health Disparities Among Minority Populations: Why Clinical Trial Recruitment Is Not Enough. *JAMA Neurol*. 2020;77(9):1063. doi:10.1001/jamaneurol.2020.1614
27. Adkins-Jackson PB, Kraal AZ, Hill-Jarrett TG, et al. Riding the merry-go-round of racial disparities in AD/DRD research. *Alzheimers Dement*. Published online July 3, 2023. doi:10.1002/alz.13359
28. Feliciano C. Shades of Race: How Phenotype and Observer Characteristics Shape Racial Classification. *American Behavioral Scientist*. 2016;60(4):390-419. doi:10.1177/0002764215613401
29. Gonzalez-Sobrinho B, Goss DR. Exploring the mechanisms of racialization beyond the black-white binary. *Ethnic and Racial Studies*. 2019;42(4):505-510. doi:10.1080/01419870.2018.1444781
30. Selod S, Embrick DG. Racialization and Muslims: Situating the Muslim Experience in Race Scholarship. *Sociology Compass*. 2013;7(8):644-655. doi:10.1111/soc4.12057
31. Bonilla-Silva E. Rethinking Racism: Toward a Structural Interpretation. *American Sociological Review*. 1997;62(3):465-480. doi:10.2307/2657316
32. Daniels J, Schulz AJ. Constructing Whiteness in Health Disparities Research. In: *Gender, Race, Class, & Health: Intersectional Approaches*. Jossey-Bass/Wiley; 2006:89-127.
33. Goosby BJ, Cheadle JE, Mitchell C. Stress-Related Biosocial Mechanisms of Discrimination and African American Health Inequities. *Annu Rev Sociol*. 2018;44(1):319-340. doi:10.1146/annurev-soc-060116-053403
34. Hobson JM, Moody MD, Sorge RE, Goodin BR. The neurobiology of social stress resulting from Racism: Implications for pain disparities among racialized minorities. *Neurobiol Pain*. 2022;12:100101. doi:10.1016/j.ynpai.2022.100101
35. Agorastos A, Chrousos GP. The neuroendocrinology of stress: the stress-related continuum of chronic disease development. *Mol Psychiatry*. 2022;27(1):502-513. doi:10.1038/s41380-021-01224-9

36. Meyer IH. Prejudice and Discrimination as Social Stressors. In: Meyer IH, Northridge ME, eds. *The Health of Sexual Minorities*. Springer US; 2007:242-267. doi:10.1007/978-0-387-31334-4_10
37. Krieger N. Measures of Racism, Sexism, Heterosexism, and Gender Binarism for Health Equity Research: From Structural Injustice to Embodied Harm-An Ecosocial Analysis. *Annu Rev Public Health*. 2020;41:37-62. doi:10.1146/annurev-publhealth-040119-094017
38. Farmer HR, Thomas Tobin CS, Thorpe RJ. Correlates of Elevated C-Reactive Protein Among Black Older Adults: Evidence From the Health and Retirement Study. Martire L, ed. *The Journals of Gerontology: Series B*. Published online February 11, 2022:gbac033. doi:10.1093/geronb/gbac033
39. Farmer HR, Wray LA, Xian Y, et al. Racial Differences in Elevated C-Reactive Protein Among US Older Adults. *J Am Geriatr Soc*. 2020;68(2):362-369. doi:10.1111/jgs.16187
40. Gouin JP, Glaser R, Malarkey WB, Beversdorf D, Kiecolt-Glaser J. Chronic stress, daily stressors, and circulating inflammatory markers. *Health Psychol*. 2012;31(2):264-268. doi:10.1037/a0025536
41. Rohleder N. Stimulation of Systemic Low-Grade Inflammation by Psychosocial Stress. *Psychosomatic Medicine*. 2014;76(3):181-189. doi:10.1097/PSY.0000000000000049
42. Geronimus AT, Hicken M, Keene D, Bound J. "Weathering" and Age Patterns of Allostatic Load Scores Among Blacks and Whites in the United States. *Am J Public Health*. 2006;96(5):826-833. doi:10.2105/AJPH.2004.060749
43. Schmeer KK, Tarrence J. Racial/Ethnic Disparities in Inflammation: Evidence of Weathering in Childhood? *J Health Soc Behav*. 2018;59(3):411-428. doi:10.1177/0022146518784592
44. Ryan J, Chaudieu I, Ancelin ML, Saffery R. Biological underpinnings of trauma and post-traumatic stress disorder: focusing on genetics and epigenetics. *Epigenomics*. 2016;8(11):1553-1569. doi:10.2217/epi-2016-0083
45. Krieger N. 11 The Science and Epidemiology of Racism and Health: Racial/Ethnic Categories, Biological Expression of Racism, and the Embodiment of Inequality—an Ecosocial Perspective. :31.
46. Hegazy SH, Thomassen JQ, Rasmussen IJ, Nordestgaard BG, Tybjaerg-Hansen A, Frikke-Schmidt R. C-reactive protein levels and risk of dementia—Observational and genetic studies of 111,242 individuals from the general population. *Alzheimer's & Dementia*. Published online February 3, 2022:alz.12568. doi:10.1002/alz.12568
47. Satizabal CL, Zhu YC, Mazoyer B, Dufouil C, Tzourio C. Circulating IL-6 and CRP are associated with MRI findings in the elderly: The 3C-Dijon Study. *Neurology*. 2012;78(10):720-727. doi:10.1212/WNL.0b013e318248e50f
48. Engelhart MJ, Geerlings MI, Meijer J, et al. Inflammatory Proteins in Plasma and the Risk of Dementia: The Rotterdam Study. *Arch Neurol*. 2004;61(5):668. doi:10.1001/archneur.61.5.668

49. Dik MG, Jonker C, Hack CE, Smit JH, Comijs HC, Eikelenboom P. Serum inflammatory proteins and cognitive decline in older persons. *Neurology*. 2005;64(8):1371-1377. doi:10.1212/01.WNL.0000158281.08946.68
50. Gabin JM, Saltvedt I, Tambs K, Holmen J. The association of high sensitivity C-reactive protein and incident Alzheimer disease in patients 60 years and older: The HUNT study, Norway. *Immunity & Ageing*. 2018;15(1):4. doi:10.1186/s12979-017-0106-3
51. Arce Rentería M, Gillett SR, McClure LA, et al. C-reactive protein and risk of cognitive decline: The REGARDS study. Ginsberg SD, ed. *PLoS ONE*. 2020;15(12):e0244612. doi:10.1371/journal.pone.0244612
52. Farmer HR, Wray LA, Haas SA. Race, Gender, and Socioeconomic Variations in C-Reactive Protein Using the Health and Retirement Study. Carr D, ed. *The Journals of Gerontology: Series B*. 2021;76(3):583-595. doi:10.1093/geronb/gbaa027
53. Khera A, McGuire DK, Murphy SA, et al. Race and gender differences in C-reactive protein levels. *J Am Coll Cardiol*. 2005;46(3):464-469. doi:10.1016/j.jacc.2005.04.051
54. Elahi FM, Miller BL. A clinicopathological approach to the diagnosis of dementia. *Nature Reviews Neurology*. 2017;13(8):457.
55. Wyss-Coray T. Inflammation in Alzheimer disease: driving force, bystander or beneficial response? *Nature medicine*. 2006;12(9):1005-1015.
56. Calsolaro V, Edison P. Neuroinflammation in Alzheimer's disease: current evidence and future directions. *Alzheimer's & dementia*. 2016;12(6):719-732.
57. Querfurth HW, LaFerla FM. Mechanisms of disease. *N Engl J Med*. 2010;362(4):329-344.
58. Schmidt R, Schmidt H, Curb JD, Masaki K, White LR, Launer LJ. Early inflammation and dementia: A 25-year follow-up of the Honolulu-Asia aging study. *Annals of Neurology*. 2002;52(2):168-174. doi:10.1002/ana.10265
59. Rea IM, Gibson DS, McGilligan V, McNerlan SE, Alexander HD, Ross OA. Age and age-related diseases: role of inflammation triggers and cytokines. *Frontiers in immunology*. 2018;9:586.
60. Engelhart MJ, Geerlings MI, Meijer J, et al. Inflammatory Proteins in Plasma and the Risk of Dementia: The Rotterdam Study. *Arch Neurol*. 2004;61(5):668-672. doi:10.1001/archneur.61.5.668
61. de Craen AJM, Gussekloo J, Vrijsen B, Westendorp RGJ. Meta-analysis of nonsteroidal antiinflammatory drug use and risk of dementia. *Am J Epidemiol*. 2005;161(2):114-120. doi:10.1093/aje/kwi029
62. Bach-Ngohou K, Nazih H, Nazih-Sanderson F, et al. Negative and independent influence of apolipoprotein E on C-reactive protein (CRP) concentration in obese adults. Potential anti-inflammatory role of apoE in vivo. *Int J Obes*. 2001;25(12):1752-1758. doi:10.1038/sj.ijo.0801833
63. APOE polymorphism and its effect on plasma C-reactive protein levels in a large general population sample | Elsevier Enhanced Reader. doi:10.1016/j.humimm.2010.01.008

64. Tang MX. The APOE- ϵ 4 Allele and the Risk of Alzheimer Disease Among African Americans, Whites, and Hispanics. *JAMA*. 1998;279(10):751. doi:10.1001/jama.279.10.751
65. Feldman MW, Lewontin RC, King MC. Race: A genetic melting-pot. *Nature*. 2003;424(6947):374-374. doi:10.1038/424374a
66. Wooten T, Brown E, Sullivan DR, et al. Apolipoprotein E (APOE) ϵ 4 moderates the relationship between c-reactive protein, cognitive functioning, and white matter integrity. *Brain Behav Immun*. 2021;95:84-95. doi:10.1016/j.bbi.2021.02.016
67. Masters CL, Bateman R, Blennow K, Rowe CC, Sperling RA, Cummings JL. Alzheimer's disease. *Nat Rev Dis Primers*. 2015;1:15056. doi:10.1038/nrdp.2015.56
68. Yasojima K, Schwab C, McGeer EG, McGeer PL. Human neurons generate C-reactive protein and amyloid P: upregulation in Alzheimer's disease. *Brain Research*. 2000;887(1):80-89. doi:10.1016/S0006-8993(00)02970-X
69. Coccaro EF, Lee R, Coussons-Read M. Cerebrospinal fluid and plasma C-reactive protein and aggression in personality-disordered subjects: a pilot study. *J Neural Transm (Vienna)*. 2015;122(2):321-326. doi:10.1007/s00702-014-1263-6
70. Crimmins E, Kim JK, McCreath H, Faul J, Weir D, Seeman T. Validation of Blood-Based Assays Using Dried Blood Spots for Use in Large Population Studies. *Biodemography and Social Biology*. 2014;60(1):38-48. doi:10.1080/19485565.2014.901885
71. Brody GH, Yu T, Miller GE, Ehrlich KB, Chen E. John Henryism Coping and Metabolic Syndrome Among Young Black Adults. *Psychosom Med*. 2018;80(2):216-221. doi:10.1097/PSY.0000000000000540
72. James SA, Keenan NL, Strogatz DS, Browning SR, Garrett JM. Socioeconomic Status, John Henryism, and Blood Pressure in Black Adults. *American Journal of Epidemiology*. 1992;135(1):59-67. doi:10.1093/oxfordjournals.aje.a116202
73. Zilioli S, Gómez JM, Jiang Y, Rodriguez-Stanley J. Childhood Socioeconomic Status and Cardiometabolic Health: A Test of the John Henryism Hypothesis in African American Older Adults. *J Gerontol A Biol Sci Med Sci*. 2021;77(2):e56-e64. doi:10.1093/gerona/glab280
74. Tejera CH, Minnier J, Fazio S, et al. High triglyceride to HDL cholesterol ratio is associated with increased coronary heart disease among White but not Black adults. *American Journal of Preventive Cardiology*. 2021;7:100198. doi:10.1016/j.ajpc.2021.100198
75. Colantonio LD, Gamboa CM, Kleindorfer DO, et al. Stroke symptoms and risk for incident coronary heart disease in the REasons for Geographic And Racial Differences in Stroke (REGARDS) study. *International journal of cardiology*. 2016;220:122-128.
76. Suzuki T, Voeks J, Zakai NA, et al. Metabolic Syndrome, C-Reactive Protein, and Mortality in U.S. Blacks and Whites: The Reasons for Geographic and Racial Differences in Stroke (REGARDS) Study. *Dia Care*. 2014;37(8):2284-2290. doi:10.2337/dc13-2059

Reviewers' comments:

Reviewer #1 (Remarks to the Author):

I thank the authors for their responses to my comments. I think that the paper has much improved since the previous time I read it. However, I do still have three comments remaining. Please find my point-by-point review below.

1. I thank the authors for clarifying their decision to dichotomize CRP. However, I still don't completely understand why the authors decided to dichotomize CRP rather than treating it as a continuous variable. Their decision to dichotomize CRP is informed by previous papers that stratified CRP to determine the risk of certain (cardiovascular) events based on CRP categories. Risk stratification can help inform the decision-making process of physicians. However, the goal of the current paper is to investigate CRP as a mechanism that may explain the association between racialization in disparities in incident dementia. In this case, effect estimates based on a continuous CRP variable may more accurately represent these mechanisms than effect estimates based on a dichotomous version of CRP for which an arbitrary cut-off was used.

2. I thank the authors for clarifying that they estimated bootstrap confidence intervals. However, CMAverse facilitates the estimation of both Percentile Bootstrap Confidence Intervals and Bias-Corrected and Accelerated Bootstrap Confidence Intervals. Could the authors clarify which of these two they estimated?

3. I appreciate that the authors added a more extensive discussion of the limitations of their study to the discussion section. However, the conclusion listed in the abstract is still rather strong and not completely supported by the results in the paper. After all, only the RERI for the mediated interaction for the analyses comparing minoritized vs. non-Hispanic White is statistically significant. None of the pure indirect effects are statistically significant. Furthermore, in general the estimates of the pure indirect effects and mediated interactions are relatively small in magnitude. I would like to see a more nuanced conclusion in the abstract to avoid misunderstanding of the results by readers who only read the abstract.

Reviewer #2 (Remarks to the Author):

The additional information in the introduction describing how racialization acts across minoritized groups to increase physiological stress and systemic inflammation, subsequently elevating risk for cognitive decline and dementia, adds to the strength of the manuscript. In general, many of my concerns were addressed by the authors; however, I remain concerned that some of the conclusions made are not sufficiently supported by the statistical evidence. The overall conclusion is that systemic inflammation mediates disparities in incident dementia among minoritized social groups; however, the mediation effect was very weak (3%) and oddly not accounted for by either the Hispanic or non-Hispanic Black groups (at least not in the 4-way mediation analysis represented in table 3). There was a higher rate of dementia incidence in Hispanic participants with higher CRP, but this result was not found for non-Hispanic Black participants. The results for non-Hispanic Black participants seem rather different than those for Hispanic participants which brings into question whether the conclusions discussed are relevant to non-Hispanic Black participants.

Although high CRP was significantly related to an increased risk for dementia in the overall sample and non-Hispanic Black participants had significantly higher odds of elevated CRP than non-Hispanic White participants, CRP did not significantly mediate or moderate the association between the non-Hispanic Black/non-Hispanic White group and dementia incidence (CIs included 0%). These results seem to suggest that CRP does not account for the dementia disparity between non-Hispanic Black and White participants, which could be an interesting finding in and of itself. John Henryism doesn't seem to be a viable explanation because one would expect that the burdens of systemic racial discrimination and the associated effortful coping that increases risk for systemic inflammation would translate into a physiological cost to health. In this analysis that cost does not

appear to be dementia. Perhaps a larger sample size is needed to identify the effect; however, a significant moderating effect of CRP was found for the association between the Hispanic/non-Hispanic White group and dementia incidence, despite the sample size of Hispanic participants being smaller than non-Hispanic Black participants.

Further, Hispanic participants with high CRP had significantly higher dementia incidence than Hispanic participants with low CRP, and in contrast to non-Hispanic Black participants, Hispanic participants did not have significantly greater odds for elevated CRP compared to non-Hispanic White participants. These results coupled with the moderation analysis seem to suggest that systemic inflammation (or confounds associated with it) may have a more deleterious effect on cognitive/brain function in individuals who self-identify as Hispanic. That finding deserves further discussion.

Some results that were statistically not significant are discussed as though they were (e.g., among minoritized participants, high CRP was associated with 1.26 (95% CI: 0.98, 1.62) times higher risk of incident dementia than low CRP [CI includes 1.0], and When decomposing the non-Hispanic Black vs non-Hispanic White disparity, we found that the mediating effect of CRP accounted for 2% (95% CI: -3%, 8%) of the disparity, and the portion attributable to the interaction accounted for 4% (95% CI: -11%, 21%) [CIs include 0]).

In the discussion about reverse causation: it is the preclinical (not prodromal) phase that is long (approximately 20 years). Based on studies of asymptomatic individuals with autosomal dominant Alzheimer's disease, which has similar pathophysiological features of late-onset AD, it is almost certain that by the time participants were identified with Alzheimer's dementia they had experienced a long preclinical period during which time amyloid and tau accumulated, possibly accompanied by neuroinflammation (DOI: 10.1056/NEJMoa1202753; <https://doi.org/10.1126/scitranslmed.3007901>).

Reviewer #3 (Remarks to the Author):

No additional comments. The authors addressed major concerns in a clear and concise manner. I look forward to seeing this study in print.

We thank the editor for the opportunity to resubmit this manuscript. The reviewers provided feedback on important aspects on the operationalization of C-reactive protein (CRP), the interpretation of the results in light of not statistically significant findings, and the observed heterogeneous effect of CRP on dementia risk between non-Hispanic Black and Hispanic participants. In response to the reviewers' feedback, we have incorporated additional sensitivity analyses, and further discussed the distinct results we observed between minoritized social groups. Our specific responses are described point by point below:

Reviewer #1

I thank the authors for their responses to my comments. I think that the paper has much improved since the previous time I read it. However, I do still have three comments remaining. Please find my point-by-point review below.

Comment # 1

I thank the authors for clarifying their decision to dichotomize CRP. However, I still don't completely understand why the authors decided to dichotomize CRP rather than treating it as a continuous variable. Their decision to dichotomize CRP is informed by previous papers that stratified CRP to determine the risk of certain (cardiovascular) events based on CRP categories. Risk stratification can help inform the decision-making process of physicians. However, the goal of the current paper is to investigate CRP as a mechanism that may explain the association between racialization in disparities in incident dementia. In this case, effect estimates based on a continuous CRP variable may more accurately represent these mechanisms than effect estimates based on a dichotomous version of CRP for which an arbitrary cut-off was used.

Answer # 1

We thank the reviewer for suggesting that effect estimates based on the continuous form of C-reactive protein (CRP) could be better suited to investigate the mechanisms implicated in racial disparities in dementia rather than an arbitrary cut-off point. We agree, the issues on dichotomizing/categorizing continuous exposures are well documented.^{1,2} To address the reviewers comment, in sensitivity models we newly tested the association between the natural logarithmic transformation of CRP (standardized) and 6-year incident dementia. We conducted multivariable Poisson regression models in the overall sample, and within each racialized social group (non-Hispanic Black, Hispanic, and non-Hispanic Black) and present those results in the **Supplemental Table 10**. In a fully adjusted model, we found that 1 standard deviation above the mean log-transformed CRP was associated with 1.06 (95%CI: 0.99, 1.14) times greater risk of dementia in the overall sample. This estimate was of slightly larger magnitude for non-Hispanic White participants (IRR = 1.07; 95%CI: 0.98, 1.16) in comparison to their non-Hispanic Black (IRR = 1.03; 95%CI: 0.89, 1.19) and Hispanic (IRR = 1.03; CI: 0.89, 1.16) counterparts. The estimated effects of CRP in its continuous form were imprecise and of lesser magnitude than those obtained when CRP was dichotomized at the 75th percentile of the sample distribution (≥ 4.73 $\mu\text{g/mL}$). Moreover, we conducted our regression-based (**Supplemental Table 11**) and g-estimation (**Supplemental Table 12**) mediation interaction decomposition analysis employing CRP in its continuous form. Since we

standardized the log-transformed CRP, we set the levels of the mediator at 0 (or the mean), and to be evaluated at 1 (or 1SD above the mean) in the CMAverse. When decomposing the minoritized vs non-Hispanic White disparity, regression-based estimates were not indicative of interaction (i.e., interaction effect virtually zero), and the mediated effect was small (percent mediated = 2%; 95%CI: 0%, 5%, p-value = 0.06) and imprecise (**Supplemental Table 11**). Similarly, when decomposing the non-Hispanic Black vs non-Hispanic White disparity, regression-based estimates were not indicative of interaction and the mediated effect was small (percent mediated = 3%; 95%CI: -2%, 9%, p-value = 0.19) and imprecise (**Supplemental Table 11**). Lastly, when decomposing the Hispanic vs non-Hispanic White disparity, regression-based estimates did not suggest mediation nor interaction. We also calculated these estimates employing randomized analogue models and using *APOE-ε4* as a variable affected by racialized social categories (exposure). We present those estimates on the **Supplemental Table 12**. Randomized analogue estimates when using CRP in its continuous form are of similar magnitude to those obtained from the previously discussed regression-based estimates (**Supplemental Table 11**).

In addition to these new continuous analyses, we have elected to keep the dichotomized analyses for several reasons. Although there are no clinical thresholds of CRP for dementia, there are well known cutoff points for cardiovascular disease, and given that dementia is frequently a mixed pathology that involves similar vascular changes to those observed in cardiometabolic diseases, employing cutoff points of CRP within the cardiovascular risk threshold might be relevant. Additionally given the high burden of cardiometabolic diseases among minoritized racial groups, our suggested cutoff point may be informative of future cognitive deterioration.

To clarify these changes to the reader we added the following tables and sentences to the manuscript.

Methods

Sensitivity Analysis

“[...] Because of the well described issues on dichotomizing continuous exposures in health science research,^{1,2} we also estimated incident rate ratios of dementia using the standardized natural logarithmic transformation of CRP as the primary predictor; and re-calculated the regression-based and g-estimation mediation-interaction decomposition models setting the mediator levels at zero (or the mean) and to be evaluated at one (or one SD above the mean) [...]”

Results

Sensitivity Analysis

*“[...] Lastly, in a fully adjusted model, we found that 1 standard deviation above the mean log-transformed CRP was associated with 1.06 (95%CI: 0.99, 1.14) times greater risk of dementia in the overall sample. This estimate was of slightly larger magnitude for non-Hispanic White participants (IRR = 1.07; 95%CI: 0.98, 1.16) in comparison to their non-Hispanic Black (IRR = 1.03; 95%CI: 0.89, 1.19) and Hispanic (IRR = 1.03; CI: 0.89, 1.16) counterparts (**Supplemental Table 10**). Nonetheless, none of these estimates achieved the statistically significant threshold*

*of 5%. When decomposing the minoritized versus non-Hispanic White disparity, the non-Hispanic Black versus the non-Hispanic White disparity, and the Hispanic versus the non-Hispanic White disparity using the natural log-transformed CRP variable in regression-based (**Supplemental Table 11**) and randomized analogue models (**Supplemental Table 12**), we did not find statistically significant evidence of interaction or mediation, suggesting that the underlying interaction and mediated effects between racialized social categories and systemic inflammation may be dependent on a particular threshold of systemic inflammation [...]*

Discussion

[...] Finally, in sensitivity models using the continuous log-transformation of CRP, we did not find statistical evidence of interaction or mediation. Further research should consider testing mediation and interaction at different cutoff points for CRP to understand if the relationship between racialization, systemic inflammation, and incident dementia is sensitive to different CRP thresholds.³ [...]

Comment # 2

I thank the authors for clarifying that they estimated bootstrap confidence intervals. However, CMAverse facilitates the estimation of both Percentile Bootstrap Confidence Intervals and Bias-Corrected and Accelerated Bootstrap Confidence Intervals. Could the authors clarify which of these two they estimated?

Answer # 2

We thank the reviewer for the opportunity to clarify the method used to calculate the confidence intervals of our decomposition components. We used the default CMAverse method: the percentile bootstrap confidence interval. To make this explicit to the reader we added the following sentence in the methods section of the manuscript.

[...] The 95% confidence intervals of our estimates were calculated using the percentile bootstrapping inference method, we performed 1000 bootstraps in each procedure and set a random seed for reproducibility purposes [...]

Comment # 3

I appreciate that the authors added a more extensive discussion of the limitations of their study to the discussion section. However, the conclusion listed in the abstract is still rather strong and not completely supported by the results in the paper. After all, only the RERI for the mediated interaction for the analyses comparing minoritized vs. non-Hispanic White is statistically significant. None of the pure indirect effects are statistically significant. Furthermore, in general the estimates of the pure indirect effects and mediated interactions are relatively small in magnitude. I would like to see a more nuanced conclusion in the abstract to avoid misunderstanding of the results by readers who only read the abstract.

Answer # 3

We thank the reviewer for this comment and the opportunity to clarify the study findings to our future audience. We agree that our pure indirect effect were not

statistically significant and have shifted the focus of the abstract conclusion to the moderating effect of the racialization process in the relationship between CRP and incident dementia. To avoid overstating our conclusions we have made the following changes to the abstract.

*“[...] **Conclusions:** Minoritized group membership modifies the relationship between systemic inflammation and incident dementia [...]”*

Reviewer #2 (Remarks to the Author):

Comment # 1

The additional information in the introduction describing how racialization acts across minoritized groups to increase physiological stress and systemic inflammation, subsequently elevating risk for cognitive decline and dementia, adds to the strength of the manuscript. In general, many of my concerns were addressed by the authors; however, I remain concerned that some of the conclusions made are not sufficiently supported by the statistical evidence. The overall conclusion is that systemic inflammation mediates disparities in incident dementia among minoritized social groups; however, the mediation effect was very weak (3%) and oddly not accounted for by either the Hispanic or non-Hispanic Black groups (at least not in the 4-way mediation analysis represented in table 3). There was a higher rate of dementia incidence in Hispanic participants with higher CRP, but this result was not found for non-Hispanic Black participants. The results for non-Hispanic Black participants seem rather different than those for Hispanic participants which brings into question whether the conclusions discussed are relevant to non-Hispanic Black participants.

Answer # 1

We thank the reviewer for their assessment of the quality and improvements made to the introduction of our manuscript. We acknowledge the reviewer's concerns on the concluding remarks of our manuscript and their apparent lack of support by the statistical evidence presented. Reviewer # 1 made a similar point, for which we made a change in the conclusions of the abstract. Instead of focusing on the mediating effect of systemic inflammation, we centered attention on the moderating effect of minoritized group membership on the relationship between systemic inflammation and incident dementia. The magnitude of the moderating effect was 14% (95CI: 1%, 27%, p-value = 0.04) and statistically significant. Although the mediating effect of systemic inflammation when decomposing the minoritized vs the non-Hispanic White disparity was very weak (3%), we have stated in the discussion that:

“[...] The slight mediation effect was expected since disparities between these groups emerge from structural forces acting differentially on groups rather than physiological processes that might be different among groups. These structural forces operate tacitly under the controlled direct effect, which represents a large proportion (88%) of the observed minoritized disparity [...]”

Our weak mediated effect may be related to the fact that, in our study, systemic inflammation was only assessed through the CRP pathway, and combining several inflammatory biomarkers may be a better strategy to capture the mediating effect of

inflammation on racial disparities in chronic health outcomes. We have further elaborated on this in the discussion section:

“[...] systemic inflammation and likely other biological responses, represent plausible mechanisms through which racism operates. In this study, we solely focused attention on a single biomarker of inflammation, but current research suggest that multiple inflammatory cytokines are related to dementia risk,^{4,5} and population-based studies that incorporate multiple inflammatory mediators as pathways to understand racialized disparities in dementia could be better suited to detect larger mediated effects. The interplay between different racialized experiences and treatment with multiple cytokine measures deserves further attention. Our results suggest that even though non-Hispanic Black participants had higher levels of CRP than their non-Hispanic White counterparts, their risk of dementia was not statistically significant when comparing those with high CRP to those with low CRP. These results indicate that in future work, studies should incorporate multiple systemic inflammatory biomarkers across diverse racialized groups to characterize the etiological role of peripheral immunity on neurodegenerative diseases [...].”

A change to the conclusion in the abstract was made as follows:

*“[...] **Conclusions:** Minoritized group membership modifies the relationship between systemic inflammation and incident dementia [...].”*

Additionally, reviewer # 2 comments that: the results for non-Hispanic Black participants seem rather different than those for Hispanic participants which brings into question whether the conclusions discussed are relevant to non-Hispanic Black participants.

We agree that there are differences in the findings among the non-Hispanic Black participants and the Hispanic participants. We hypothesize that selection bias issues among non-Hispanic Black participants may explain these findings and require further examination. If systemic inflammation is causally related to chronic disease development and earlier mortality occurs in participants racialized as non-Hispanic Black, it can be the case that multimorbid conditions and early mortality are events precluding Black participants from developing dementia. For instance, the non-Hispanic Black participants that remain in the study cohort may differ from those of the general population in their susceptibility to chronic disease development. In the discussion we have stated that:

“[...] Moreover, in stratified mediation models, we did not observe statistically significant mediated or moderated effects when comparing the non-Hispanic Black versus non-Hispanic White disparity than when comparing the minoritized disparity. We attribute these findings to the null association between CRP and incident dementia among non-Hispanic Black participants, for which we have other possible explanation. We hypothesize that the higher levels of CRP found in non-Hispanic Black participants are characteristic of a chronic stress response that results from persistent experiences with structural racism.⁶⁻⁸ Therefore, chronic systemic inflammation may predispose Black participants for other competing events such as diabetes, cardiovascular disease, stroke, and premature death,⁹⁻¹¹ which in turn may affect Black participants' likelihood of retention during the study period. Although our models controlled for confounding bias by these potential competing events, we did

not account for selection bias issues in our analysis, and future research should inform how differential loss to follow-up affects the relationship between systemic inflammation and dementia in Black participants [...]

Comment # 2

Although high CRP was significantly related to an increased risk for dementia in the overall sample and non-Hispanic Black participants had significantly higher odds of elevated CRP than non-Hispanic White participants, CRP did not significantly mediate or moderate the association between the non-Hispanic Black/non-Hispanic White group and dementia incidence (CIs included 0%). These results seem to suggest that CRP does not account for the dementia disparity between non-Hispanic Black and White participants, which could be an interesting finding in and of itself. John Henryism doesn't seem to be a viable explanation because one would expect that the burdens of systemic racial discrimination and the associated effortful coping that increases risk for systemic inflammation would translate into a physiological cost to health. In this analysis that cost does not appear to be dementia. Perhaps a larger sample size is needed to identify the effect; however, a significant moderating effect of CRP was found for the association between the Hispanic/non-Hispanic White group and dementia incidence, despite the sample size of Hispanic participants being smaller than non-Hispanic Black participants.

Further, Hispanic participants with high CRP had significantly higher dementia incidence than Hispanic participants with low CRP, and in contrast to non-Hispanic Black participants, Hispanic participants did not have significantly greater odds for elevated CRP compared to non-Hispanic White participants. These results coupled with the moderation analysis seem to suggest that systemic inflammation (or confounds associated with it) may have a more deleterious effect on cognitive/brain function in individuals who self-identify as Hispanic. That finding deserves further discussion.

Answer comment # 2

We thank the reviewer for this insightful comment and agree that when analyzing the relationship between high CRP levels and incident dementia, the heterogeneous results between non-Hispanic Black and Hispanic participants were puzzling. To clarify to the reader the observed heterogeneous effect of CRP on dementia between non-Hispanic Black and Hispanic participants, we have added the following remark:

"[...] The substantial heterogeneity in the relationship between high CRP and dementia risk across the distinct racialized experiences and treatment of minoritized social groups shows that Hispanic participants are more susceptible to the effect of high CRP levels on cognitive health than non-Hispanic Black participants. This suggests that unique racialized processes link biological pathways to health outcomes. Studies should further explore how diverse racialized experiences (i.e., immigration, segregation, unemployment, underemployment) influence inflammatory-related pathways and their relation to cognitive health in minoritized populations. It is noteworthy to mention that when comparing the minoritized status of non-Hispanic Black and Hispanic participants together to the most privileged social position of non-Hispanic White participants, the effect of systemic inflammation on dementia risk

was moderated by the participants' position on this binary spectrum, demonstrating that on average minoritized group membership influence inflammatory pathways and brain health [...]

Comment # 3

Some results that were statistically not significant are discussed as though they were (e.g., among minoritized participants, high CRP was associated with 1.26 (95% CI: 0.98, 1.62) times higher risk of incident dementia than low CRP [CI includes 1.0], and When decomposing the non-Hispanic Black vs non-Hispanic White disparity, we found that the mediating effect of CRP accounted for 2% (95% CI: -3%, 8%) of the disparity, and the portion attributable to the interaction accounted for 4% (95% CI: -11%, 21%) [CIs include 0]).

Answer # 3

We thank the reviewer for the opportunity to clarify the statistically significant threshold of our estimates and have made the following changes when describing our associations.

“[...] For example, among minoritized participants, high CRP was associated with 1.26 (95% CI: 0.98, 1.62) times higher risk of incident dementia than low CRP, although this finding was not statistically significant. Similarly, the risk of 6-year incident dementia for non-Hispanic White participants with high CRP was 1.19 (95% CI: 0.98, 1.45) times higher than those with low CRP, but this finding was not statistically significant. However, among Hispanic participants, high CRP was associated with 1.85 (95% CI: 1.27, 2.70) times higher risk of dementia than low CRP. Among non-Hispanic Black participants, the association between CRP and incident dementia was null (IRR: 1.00; 95%CI: 0.72, 1.37) (Table 2) [...]”

“[...] When decomposing the non-Hispanic Black vs non-Hispanic White disparity, we found that the mediating effect of CRP accounted for 2% (95% CI: -3%, 8%) of the disparity, and the portion attributable to the interaction accounted for 4% (95% CI: -11%, 21%), while neither of these estimates was statistically significant (Table 3 & Supplemental Figure 3) [...]”

“[...] When decomposing the non-Hispanic Black vs non-Hispanic White disparity, we found that the mediating effect of CRP accounted for 2% (95% CI: -3%, 7%) of the disparity, and the portion attributable to the interaction accounted for 5% (95% CI: -12%, 23%), but these estimates were not statistically significant (Supplemental Table 6 & Supplemental Figure 3) [...]”

Comment # 4

In the discussion about reverse causation: it is the preclinical (not prodromal) phase that is long (approximately 20 years). Based on studies of asymptomatic individuals with autosomal dominant Alzheimer's disease, which has similar pathophysiological features of late-onset AD, it is almost certain that by the time participants were identified with Alzheimer's dementia they had experienced a long preclinical period during which time amyloid and tau accumulated, possibly accompanied by neuroinflammation (DOI: 10.1056/NEJMoa1202753;

<https://doi.org/10.1126/scitranslmed.3007901>).

Answer # 4

We thank the reviewer for their comment, we have now corrected the word prodromal for preclinical and referenced their suggested source.

“Lastly, during the preclinical phase of Alzheimer’s disease (20 years) there are changes in cerebrospinal fluid concentrations of Amyloid- β 42 and other inflammatory biomarkers that are predictive of disease onset.^{15,16} In large observational studies like the Health and Retirement Study, participants are routinely screened for changes in their cognitive function; clinical impairment debuts with changes in cognitive tests scores starting approximately 6 years before symptoms onset.¹⁵ We used a 6-year follow-up period to estimate incident dementia, but it is plausible that participants classified as incident cases may have experienced a long preclinical period with changes in brain anatomy and neuro-inflammatory biomarkers.^{15,16} Because of the colocalization of CRP with amyloid- β plaques in brain parenchyma, and the correlation between CRP cerebrospinal fluid concentrations and peripheral levels, our results may be susceptible to reverse causation.^{17–19}”

Reviewer #3 (Remarks to the Author):

Comment # 1

No additional comments. The authors addressed major concerns in a clear and concise manner. I look forward to seeing this study in print.

Answer # 1

We thank the reviewer for their kind assessment of our manuscript and appreciate their opportunity to present our science to a broader audience.

References

1. Weinberg CR. How Bad Is Categorization? *Epidemiology*. 1995;6(4):345-347. Accessed January 17, 2024. <https://www.jstor.org/stable/3702077>
2. The Cost of Dichotomization. doi:10.1177/014662168300700301
3. VanderWeele TJ, Chen Y, Ahsan H. Inference for Causal Interactions for Continuous Exposures under Dichotomization. *Biometrics*. 2011;67(4):1414-1421. doi:10.1111/j.1541-0420.2011.01629.x
4. Li C, Stebbins RC, Noppert GA, et al. Peripheral immune function and Alzheimer’s disease: a living systematic review and critical appraisal. *Mol Psychiatry*. Published online December 15, 2023:1-11. doi:10.1038/s41380-023-02355-x
5. Zhang YR, Wang JJ, Chen SF, et al. Peripheral immunity is associated with the risk of incident dementia. *Mol Psychiatry*. 2022;27(4):1956-1962. doi:10.1038/s41380-022-01446-5
6. Adkins-Jackson PB, Chantarat T, Bailey ZD, Ponce NA. Measuring Structural Racism: A Guide for Epidemiologists and Other Health Researchers. *American Journal of Epidemiology*. 2022;191(4):539-547. doi:10.1093/aje/kwab239

7. Hobson JM, Moody MD, Sorge RE, Goodin BR. The neurobiology of social stress resulting from Racism: Implications for pain disparities among racialized minorities. *Neurobiol Pain*. 2022;12:100101. doi:10.1016/j.ynpai.2022.100101
8. Meyer IH. Prejudice and Discrimination as Social Stressors. In: Meyer IH, Northridge ME, eds. *The Health of Sexual Minorities*. Springer US; 2007:242-267. doi:10.1007/978-0-387-31334-4_10
9. Tejera CH, Minnier J, Fazio S, et al. High triglyceride to HDL cholesterol ratio is associated with increased coronary heart disease among White but not Black adults. *American Journal of Preventive Cardiology*. 2021;7:100198. doi:10.1016/j.ajpc.2021.100198
10. Colantonio LD, Gamboa CM, Kleindorfer DO, et al. Stroke symptoms and risk for incident coronary heart disease in the REasons for Geographic And Racial Differences in Stroke (REGARDS) study. *International journal of cardiology*. 2016;220:122-128.
11. Suzuki T, Voeks J, Zakai NA, et al. Metabolic Syndrome, C-Reactive Protein, and Mortality in U.S. Blacks and Whites: The Reasons for Geographic and Racial Differences in Stroke (REGARDS) Study. *Dia Care*. 2014;37(8):2284-2290. doi:10.2337/dc13-2059
12. Brody GH, Yu T, Miller GE, Ehrlich KB, Chen E. John Henryism Coping and Metabolic Syndrome Among Young Black Adults. *Psychosom Med*. 2018;80(2):216-221. doi:10.1097/PSY.0000000000000540
13. James SA, Keenan NL, Strogatz DS, Browning SR, Garrett JM. Socioeconomic Status, John Henryism, and Blood Pressure in Black Adults. *American Journal of Epidemiology*. 1992;135(1):59-67. doi:10.1093/oxfordjournals.aje.a116202
14. Zilioli S, Gómez JM, Jiang Y, Rodriguez-Stanley J. Childhood Socioeconomic Status and Cardiometabolic Health: A Test of the John Henryism Hypothesis in African American Older Adults. *J Gerontol A Biol Sci Med Sci*. 2021;77(2):e56-e64. doi:10.1093/gerona/glab280
15. Masters CL, Bateman R, Blennow K, Rowe CC, Sperling RA, Cummings JL. Alzheimer's disease. *Nat Rev Dis Primers*. 2015;1:15056. doi:10.1038/nrdp.2015.56
16. Fagan AM, Xiong C, Jasielc MS, et al. Longitudinal Change in CSF Biomarkers in Autosomal-Dominant Alzheimer's Disease. *Science Translational Medicine*. 2014;6(226):226ra30-226ra30. doi:10.1126/scitranslmed.3007901
17. Hegazy SH, Thomassen JQ, Rasmussen IJ, Nordestgaard BG, Tybjaerg-Hansen A, Frikke-Schmidt R. C-reactive protein levels and risk of dementia—Observational and genetic studies of 111,242 individuals from the general population. *Alzheimer's & Dementia*. Published online February 3, 2022:alz.12568. doi:10.1002/alz.12568
18. Yasojima K, Schwab C, McGeer EG, McGeer PL. Human neurons generate C-reactive protein and amyloid P: upregulation in Alzheimer's disease. *Brain Research*. 2000;887(1):80-89. doi:10.1016/S0006-8993(00)02970-X
19. Coccaro EF, Lee R, Coussons-Read M. Cerebrospinal fluid and plasma C-reactive protein and aggression in personality-disordered subjects: a pilot study. *J Neural Transm (Vienna)*. 2015;122(2):321-326. doi:10.1007/s00702-014-1263-6

REVIEWERS' COMMENTS:

Reviewer #1 (Remarks to the Author):

I thank the authors for their responses to my comments. I appreciate that the authors added additional analyses with CRP as a continuous variable. At this point, I have only one remaining minor comment.

1. In the revised methods section, the authors state that they set the mediator levels to zero and one. For readers less familiar with mediation analysis, it would be helpful to clarify that the mediator levels are set to zero and one for the estimation of the controlled direct effect.

Reviewer #2 (Remarks to the Author):

I have reviewed the manuscript by Dr. Tejera and colleagues. My previous concerns have been addressed, and I have no additional comments. I look forward to seeing the manuscript in print.

Reviewer #1:

Remarks to the Author:

I thank the authors for their responses to my comments. I appreciate that the authors added additional analyses with CRP as a continuous variable. At this point, I have only one remaining minor comment.

1. In the revised methods section, the authors state that they set the mediator levels to zero and one. For readers less familiar with mediation analysis, it would be helpful to clarify that the mediator levels are set to zero and one for the estimation of the controlled direct effect.

Response: We thank the reviewer for their assessment of our manuscript and appreciate their clarification comment with regards to the operationalization of a continuous mediator to estimate the controlled direct effect. To address this comment, we included the following clarifying language in the methods section of the manuscript. "[...] We also estimated incident rate ratios of dementia using the standardized natural logarithmic transformation of CRP as the primary predictor; and re-calculated the regression-based and g-estimation mediation-interaction decomposition models setting the mediator levels at zero (or the mean) and to be evaluated at one (or one SD above the mean) for the estimation of the controlled direct effect. [...]"

Reviewer #2:

Remarks to the Author:

I have reviewed the manuscript by Dr. Tejera and colleagues. My previous concerns have been addressed, and I have no additional comments. I look forward to seeing the manuscript in print.

Response: We thank the reviewer for their previous comments and their kind assessment of our manuscript.